# Perinatal Dietary Polyunsaturated Fatty Acids in Brain Development, Role in Neurodevelopmental Disorders

**DOI:** 10.3390/nu13041185

**Published:** 2021-04-02

**Authors:** Maud Martinat, Moïra Rossitto, Mathieu Di Miceli, Sophie Layé

**Affiliations:** Laboratoire NutriNeuro, UMR INRAE 1286, Bordeaux INP, Université de Bordeaux, 146 Rue Léo Saignat, CEDEX, 33076 Bordeaux, France; maud.martinat@inrae.fr (M.M.); moira.rossitto@inrae.fr (M.R.); mathieu.di-miceli@inrae.fr (M.D.M.)

**Keywords:** n-3 PUFAs, n-6 PUFAs, neurodevelopment, neuroinflammation, ASD, ADHD, schizophrenia, DHA, EPA, FADS, ELOVL, polymorphism, microglia, sex differences, placenta

## Abstract

n-3 and n-6 polyunsaturated fatty acids (PUFAs) are essential fatty acids that are provided by dietary intake. Growing evidence suggests that n-3 and n-6 PUFAs are paramount for brain functions. They constitute crucial elements of cellular membranes, especially in the brain. They are the precursors of several metabolites with different effects on inflammation and neuron outgrowth. Overall, long-chain PUFAs accumulate in the offspring brain during the embryonic and post-natal periods. In this review, we discuss how they accumulate in the developing brain, considering the maternal dietary supply, the polymorphisms of genes involved in their metabolism, and the differences linked to gender. We also report the mechanisms linking their bioavailability in the developing brain, their transfer from the mother to the embryo through the placenta, and their role in brain development. In addition, data on the potential role of altered bioavailability of long-chain n-3 PUFAs in the etiologies of neurodevelopmental diseases, such as autism, attention deficit and hyperactivity disorder, and schizophrenia, are reviewed.

## 1. Introduction

Lipids are critical biochemical components of the brain and are essential for proper brain functions. The lipid composition of the brain is unique and exceedingly diverse. Aberrant brain lipid composition, metabolism, and signaling are associated with neurodevelopmental, neuropsychiatric, and neurodegenerative diseases [1]. Polyunsaturated fatty acids (PUFAs) from the n-6 and n-3 families are lipids that rely on dietary supply and are considered crucial for brain development [1,2,3]. The PUFAs found in nuts, seeds, and certain vegetables, such as alpha-linolenic acid (ALA, n-3) and linoleic acid (LA, n-6), are essential to humans, and mammals in general, because they cannot synthesize them [4,5]. These PUFAs depend on nutritional intake, with ALA and LA found in very distinct sources. Indeed, ALA is found in flaxseed (linseed), English walnut, hemp seed, and chia, while LA is found in soybean oil, safflower oil, and corn oil. Once consumed, both ALA and LA are metabolized into long-chain (LC) PUFAs; ALA to eicosapentaenoic (EPA, n-3) and docosahexaenoic acid (DHA, n-3), and LA to arachidonic acid (ARA, n-6) [1]. Nutritional sources of EPA and DHA are found in fat fishes, seafood, and marine microalgae or animal products fed with ALA and/or DHA (such as eggs, fish, and livestock).

DHA and ARA are particularly aggregated in the brain during the developmental period, while the accretion of EPA in the brain is negligible [6,7], which is discussed below. Importantly, PUFAs from the mother represent the only source of LC n-3 PUFAs for the fetus [8]. According to international agencies such as the National Health Security Agency (ANSES), the European Scientific Committee on Food (ESCF), and the International Society for the Study of Fatty Acids and Lipids (ISSFAL), the recommendations for LC n-3 PUFA intake for optimal development is around 500 mg/day of EPA and DHA [9]. Based on clinical studies, the dose for EPA + DHA supplementation generally recommended to pregnant women can range from 200 to 1000 mg/day [10]. Of note, health organizations recommend increased consumption of LC n-3 PUFAs-rich marine products. Nevertheless, there is no consensus on the exact dose of EPA and DHA required during gestation and lactation for optimal brain development. In addition, the origin, safety, and sustainable supply of the marine sources have raised concerns. Alternative sustainable sources, such as microalgae-based LC n-3 PUFAs, are investigated [11]. Overall, there is a need for a better understanding of how EPA and DHA contribute to brain development to define specific recommendations for the fetal and post-natal brain. Conversion of LA and ALA to LC PUFAs, respectively ARA-EPA and DHA, is mediated by successive steps of desaturation and elongation. Importantly, both LA and ALA compete for the same enzymes in their biosynthesis (Figure 1). The enzymes involved in desaturation and elongation are rate-limiting enzymes. Delta-5 desaturase (D5D) and delta-6 desaturase (D6D) are respectively encoded by *fatty acid desaturase 1* (*fads1*) and *fatty acid desaturase 2* (*fads2*). Elongase 2 (Elovl2) and elongase 5 (Elovl5) are respectively encoded by *elovl2* and *elovl5* [12]. The last step for DHA synthesis involves translocation of 24:6 n-3 from the endoplasmic reticulum to peroxisomes, where two carbons are removed to form LC n-3 PUFAs [13]. Of note, Elovl4, an enzyme that mediates elongation of LC PUFAs and saturated fatty acids (SFAs) to form very LC PUFAs and very LC SFAs (28 to 38 carbon chain length), has been detected in the brain [14,15]. EPA is preferred as a substrate for elongation to very LC PUFAs over ARA and DHA [16]. However, the main Elovl4-derived products are very LC SFAs, with very LC PUFAs being only present in traces in the brain [17]. More studies are needed for the link between Elovl4 and very LC PUFAs in the brain. Desaturases have a higher affinity for n-3 PUFAs; however, due to the general higher consumption of LA, desaturation of n-6 PUFAs is greater. Importantly, genetic variations in *fads* and *elovl* genes affect the status of LC PUFAs independently of dietary effects [18]. Indeed, more attention is now given to polymorphisms in these genes, especially during brain development [19]. Single Nucleotide Polymorphisms (SNPs) within these genes can be associated with variations of LC PUFAs in the plasma (Figure 1), as demonstrated before [20]. SNPs in *fads* have been shown to be associated with various health outcomes (including markers of metabolic syndrome) and the plasma lipid profile in children, as well as neurodevelopmental outcomes in breastfed children [21]. The authors also found that breastmilk PUFAs content was inversely correlated to the production of inflammatory factors such as cytokine in infants [21]. Of note, low dietary intake of LC n-3 PUFAs leads to additional EPA and DHA deficiency in subjects with *fads* and *elovl* gene polymorphisms. As an example, some recent data pinpoint that low DHA status in vegetarian women may adversely affect the development of their children [22]. On the contrary, it is not clear whether consumption of diets rich in LA could be detrimental. Indeed, a clinical study conducted in 2001 on mothers supplemented with cod liver oil (enriched in LC n-3 PUFAs) or corn oil (enriched in LC n-6 PUFAs) showed no significant differences regarding pregnancy outcomes and cognitive development or growth of children [23]. Until recently, the capacity of metabolization of PUFA precursors into LC PUFAs was believed to be uniform in individuals and populations. The discovery that European and African populations carry different forms of *fads* alleles may partially explain the differences between blood levels of LC PUFAs in these populations (reviewed in [24]). The geographical differences in *fads* alleles are probably linked to specific selection in the European and African populations due to different food habits [25]. This knowledge, in combination with dietary evaluation, may help to refine dietary recommendation target more precisely population for personalized dietary supplementation in pregnant/lactating women and children at risk of altered level of LC PUFAs.

In addition to their crucial role in the structure and function of cell membranes, n-3 and n-6 PUFAs are substrates for the production of several signaling molecules involved in the physiological function of the cells, especially in the brain [1,7]. Indeed, EPA, DHA, and ARA are hydrolyzed by the specific phospholipase A2 (PLA2) and can be further metabolized by cyclooxygenases (COXs), lipooxygenases (LOXs), and the cytochrome P450 system into eicosanoids (prostaglandins, leukotrienes, and hydroxyeicosatetraenoic acids from ARA) or docosanoids (resolvins, protectins, maresins, and neuroprotectin D1 from EPA and DHA), which are named oxylipins. n-3 and n-6 PUFAs-derived oxylipins are crucial to homeostatic functions but often have opposite activities, with ARA-derived eicosanoids being pro-inflammatory and DHA/EPA-derived being anti-inflammatory and pro-resolutive [26,27,28,29]. Their role in brain inflammation has been reviewed recently [7]; however, data on the developing brain are rather scarce and are discussed in the present review.

The massive changes in dietary habits from a balanced n-3/n-6 PUFAs ratio to excess n-6 PUFAs in Western societies across the twentieth century raises the question of its impact on brain development and its contribution to neurodevelopmental disorders and later life brain health [2]. Concerning n-3 PUFAs and infant brain development and mental health, the understanding of gene/nutrient interactions may be particularly important for the design of specific nutritional strategies, so-called precision nutrition [24,30].

In this review, we will discuss how maternal and child nutrition and polymorphism of PUFA metabolism genes influence their aggregation and activity in the developing brain, considering the placenta transfer to the in utero developing brain. Specific attention was given to gender. Then, we will discuss the potential role of these PUFAs in neurodevelopmental disorders, focusing on autism spectrum disorders (ASD), attention deficit and hyperactivity disorder (ADHD), and schizophrenia.

## 2. PUFAs and Brain Development

### 2.1. Accretion of PUFAs in the Developing Brain

Brain cell membranes are particularly rich in lipids (50–70%) [31], with LC PUFAs DHA and ARA being the highest species detectable, even if other fatty acids are also detected (such as monounsaturated and saturated fatty acids). As a result, the human prefrontal cortex contains around 30% of PUFAs, with 14% of DHA and 9% of ARA [32]. In rodents, the fatty acid composition of the brain is similar to humans; 12% of DHA and 10% of ARA [33]. Fatty acids are esterified in several type of phospholipids: phosphatidylcholine (PC, 42–44%), phosphatidylethanolamine (PE, 36–40%) phosphatidylserine (PS, 11–13%), and phosphatidylinositol (PI, 2–3%) [31]. DHA is mostly esterified in PE and PS, whereas ARA is esterified in PC and PI [1,34,35]. Striking differences can be found between the white and the grey matter, with DHA and ARA levels being lower in the white matter compared to the grey matter [36]. These differences could be due to the accumulation of DHA in synaptosomes and ARA in vascular cell membranes of the brain, which is suggested to serve as a depot for ARA [36,37]. Besides, the lipidic composition of the myelin sheath could also explain these differences since cholesterol is abundant in myelinated axons, while the grey matter is highly enriched in PUFAs [38,39]. High levels of DHA in synaptosomes are consistent with previous data, demonstrating that DHA promotes neurite outgrowth and synaptogenesis [40,41], which is discussed below.

Brain accumulation of both n-3 and n-6 PUFAs starts during gestation; a process often referred to as “accretion”. As previously noted, the pre and post-natal increase in DHA depends on maternal and infant DHA intake [42,43]. DHA accumulation in the brain is massive, with a nearly 30-fold increase in the first two years of life [44]. In humans, DHA accretion begins during the third trimester of pregnancy [45], while accretion during the first or second trimester is low [46]. In the third trimester, accretion of DHA and ARA is substantial, while accretion of the precursors LA and ALA is minimal [45] and EPA remains poorly aggregated. In humans, exponential accretion of LC n-3 and n-6 PUFAs was observed between gestational weeks 26 and 43, while post-natal LC PUFAs accretion is rather steady [47]. In humans, DHA accretion reaches 40 mg/day in the fetus during the final four weeks of gestation [48]. The high DHA demand for the fetal brain is sustained by the mother, thanks to the increased DHA synthesis from ALA and mobilization of maternal DHA stores from adipose tissue [49,50]. For comparison, in rodents, DHA accumulates in the brain during the first three weeks of neonatal life with a 10-fold increase [51].

There is little data on the rate of accretion of LC PUFAs in different brain cell types (neurons, astrocytes, microglia, and oligodendrocytes) during brain development. This knowledge would be of high interest as neurons and glial cells do not proliferate or mature at the same development stage. In addition, the role of DHA/ARA in the specificity of development is still unknown. As previously mentioned, DHA accumulates in synaptosomes and regulates the functions of synaptic proteins [40,52,53]. DHA is also found in astrocytes [54], microglia [55], and oligodendrocyte membranes [56,57]. Interestingly, recent data show that DHA in the membrane of glial cells regulates their activity, the formation of syncytium in astrocytes, neuroinflammation in microglia [7], or myelin formation in oligodendrocytes [58]. Of note, EPA is found in microglia [55,59], suggesting that these cells could play an important role in mediating the neuroprotective effect of EPA dietary supplementation in mood disorders [60]. However, the precise role of DHA, EPA, and ARA accretion in these cells during development and the impact of altered accretion on glial cell development remains to be investigated for both term and preterm neonates. Indeed, children born preterm present major alterations of DHA and ARA accretion, as they normally accumulate during the third trimester of gestation, thus exposing preterm neonates to a deficit in LC n-3 PUFAs [61]. Recent data revealed a detrimental role of ARA in the microglia phagocytic activity in the spine, a crucial mechanism of developmental neuronal network wiring [59]. More studies are needed to better understand the type of PUFAs and the dynamic of PUFAs accretion in the different glial cells during development.

Overall, the massive increase in DHA and ARA in the third trimester of the developing brain coincides with the growth spurt characterized by an intensive neuron outgrowth and synaptic wiring [44]. Acquisition of new knowledge is promising for a better understanding of the importance of the optimal nutritional amount of n-3 PUFAs during gestation and lactation and young infants in both humans and rodents.

### 2.2. Needs of LC PUFAs to the Pre and Post-Natal Developing Brain: Transport and Dietary Maternal Supply

#### 2.2.1. Pre-Natal Maternal Supply of LC PUFAs to the Developing Brain via the Placenta and the Blood-Brain Barrier

As previously reported, important concentrations of LC PUFAs, particularly ARA and DHA, are accumulated in the brain during early life to support the growth and development of the brain [62]. Numerous studies have demonstrated the importance of LC n-3 and n-6 PUFAs intake by the mother for adequate intake in the fetus and new-born. This contribution is made by maternal dietary intake, which is transferred through the placenta during gestation [63], then by milk during lactation [64]. Particular attention has been paid in the last years to DHA dietary intake in pregnant women to promote optimal brain development of the fetus [65], as the fetus is not able to produce its own DHA and is therefore dependent on maternal supply [66]. According to the World Health Organization, pregnant women should consume at least 200 mg/day of DHA. This amount of DHA intake is poorly met in the general population [67], raising the question of supplementation in DHA of pregnant women, especially the ones at risk for premature birth [68]. An omega-3 Index of 8–10% (DHA plus EPA levels in erythrocytes) is a target range to avoid risks and complications during pregnancy and lactation, such as premature birth [69]. Two major randomized controlled trials conducted in Australia and the United States, respectively DHA to Optimize Mother Infant Outcome (DOMInO) and Kansas DHA Outcomes Study (KUDOS), revealed that supplementation of mothers with 800 and 600 mg of DHA/day significantly reduced the number of preterm births [70] with a poor effect on the neurodevelopment of children [71]. However, a large multicentric trial (ORIP) aiming at studying the effect of prenatal LC n-3 PUFAs supplementation (800 mg of DHA + 100 mg of EPA) on the incidence of preterm birth [72] did not observe a beneficial effect on length of pregnancy [73]. Maternal plasma level of EPA + DHA below 2% in the first trimester of gestation could be a standard evaluation to perform dietary LC n-3 PUFA supplementation [74]. This reinforces the need for monitoring LC PUFAs during pregnancy to design appropriate dietary supplementation and prevent the adverse effects of insufficient supply of these fatty acids to the developing brain.

Maternal lipids are transferred to the fetus across the placenta via free or specific transport of unesterified fatty acids [75]. Indeed, during the third trimester of gestation, PUFAs are transferred from mother to fetus via the placenta, thanks to fatty acid translocase (FAT/CD36) and fatty acid transport protein (FATP), both present at the placental membrane [76]. These transporters are located on both microvillus and basal membranes (maternal and fetal sides, respectively) of human placental cells and can transport free fatty acids in both directions. The placental membrane fatty acid-binding protein (p-FABP_pm_) is only located on the microvillus membrane and can transport ARA and DHA from maternal plasma to the fetus. The presence of p-FABP_pm_ exclusively on the maternal side may favor the unidirectional flow of LC PUFAs from the mother to the fetus. [76,77,78]. The cytosolic form of FABP is also detected in both primary human trophoblasts and human placental choriocarcinoma (BeWo) cells, two models of in vitro embryonic culture [76,79,80], as well as in murine placenta [81,82,83].

Regarding DHA, which is crucial for optimal brain development, free DHA is transferred from the mother to the offspring via the placenta [84,85,86,87]. Several plasma pools are a major source for free DHA [88]. Indeed, plasma DHA is esterified in several lipids, phospholipids (PL), lysophospholipids, cholesteryl-esters, and triacyl-glycerol (TG). Hydrolysation of chylomicrons rich in TG by lipoprotein lipases (LPL) is an important source of free DHA found in the blood. During gestation, a free DHA maternal pool is a crucial source for placental and fetal DHA uptake and accretion [50,89].

The blood-brain barrier (BBB) limits the entry of blood cells, neurotoxic plasma molecules, pathogens and regulates the delivery of metabolites and essential nutrients to the brain, including LC PUFAs. Several transport systems regulating nutrients entry are expressed by endothelial cells of the BBB, such as glucose, specific amino-acids, vitamins, fatty acids, and DHA transporters [90]. Free DHA and lysophosphatidylcholine (LPC)-DHA are the primary sources of brain DHA [88]. The brain uptake of these forms of DHA relies on several mechanisms, including free passage and specific transporters on the BBB [1]. Recently, the major facilitator superfamily domain-containing protein 2a (Mfsd2a), which is located at the BBB, has been identified as a transporter of LPC-DHA, especially during the post-natal period with an impact on neuronal arborization [91,92]. This transporter is key to the brain amount of DHA [92] and the maintenance of the BBB while also inhibiting the transport of toxic molecules [93,94]. During development, pericytes are necessary to Mfsd2a expression and BBB formation [90]. Indeed, Mfsd2a confers to the endothelial cell of the BBB a specific lipid composition that is crucial to its permeability [94]. In addition, LPC-DHA, through Mfsd2a, represses *de novo* lipogenesis, with a profound effect on brain cell membrane composition in phospholipids during development [92]. Mfsd2a has also been identified in the human placenta, allowing the transport of LPC-DHA from mother to the fetus [95,96]. Very interestingly, deletion or total mutations of *mfsd2a* are responsible in humans of microencephaly and hypomyelination, further reinforcing its importance in brain shaping during development, possibly through DHA brain supply [58,97,98]. Of note, a recent study reports that Zika infection during pregnancy, a flavivirus that triggers fetal brain defect, including hydrocephalus, and alters Mfsd2a and DHA levels in the developing brain [99]. In mice, the genetic deletion of *mfsd2a* reduces DHA brain level and induces microcephaly [91,92]. Thanks to this KO in mice, the authors demonstrated that the expression of Mfsd2a is regulated by sterol regulatory element-binding proteins (Srebps) [92]. Srebp, a transcription factor, exists in three isoforms: Srebp-1a and -1c, both regulating genes required for lipogenesis, and Srebp-2, regulating genes in the metabolism of cholesterol with the two isoforms -1c and -2 being predominant in the brain [100]. A high level of EPA/DHA in the diet in mice induces a decrease in the expression of Srebp-1 in the brain [101]. Srebp-1 expression is significantly reduced in the dysbindin-1 KO mouse model of schizophrenia and post-mortem brain tissue from patients with schizophrenia [102]. Recently, a team showed in mice that KO of Srebp-1c induced an alteration of GABAergic transmission, leading to symptoms similar to schizophrenia: hyperactivity, depression-like symptoms, and social deficits [103]. All these results show that SREBP-1 could play a role in synaptic plasticity and transmission via the regulatory loop with n-3 PUFAs.

#### 2.2.2. Post-Natal Maternal Supply of LC PUFAs to the Developing Brain via Breastmilk or Infant Formula

As already mentioned, endogenous synthesis of LC PUFAs from their precursors (LA and ALA) is limited during infancy [104]. Consequently, post-natal brain uptake of LC PUFAs relies on maternal milk or formulas [44]. In the maternal circulation, concentrations of LC n-3 and n-6 PUFAs (mostly DHA and ARA) are lower than in the new-born, while concentrations of precursors (ALA and LA) are higher in the maternal circulation [105]. Interestingly, *post-mortem* studies report that breastfed infants have greater amounts of DHA in the cerebral cortex than formula-fed infants without ARA or DHA supplementation [106]. Breastfed infants also present higher *post-mortem* DHA levels in both erythrocytes and the cortex but not in the retina [42]. Here we will focus on post-natal LC PUFAs supply via maternal breastmilk or formulas.

According to the World Health Organization, only 41% of infants under the age of six months are breastfed. Breastfeeding is the most adequate source of nutrients for infants and provides protection against child infections, increases intelligence, and protects against overweight and diabetes due to the presence of antibodies and lipids in maternal milk [107]. Depending on the age of the new-born, three different phases of milk production are distinguished, with three different milk compositions. First, the breast produces the colostrum, beginning on the third trimester of pregnancy until few days after birth. These fluids allow physiological adaptation of the new-born to extra uterine life. After transitional milk (until two weeks after delivery), the breast produces mature milk, which provides high amounts of lipids to the new-born [108]. During the first year of life, breast-milk LC PUFAs concentrations remain rather stable. ARA content is equivalent to 1% of milk total lipids in the colostrum and 0.5% within mature milk, which accounts for 14–15 mg/dL. In parallel, DHA content is equivalent to 0.5% in the colostrum and 0.25% in mature milk, accounting for 7–8 mg/dL [109]. The fatty acid patterns within maternal milk are greatly influenced by maternal lipid intake [110,111]. In fact, lactating mothers with dietary fish oil supplementation (rich in EPA and DHA) displayed increased levels of n-3 PUFAs in breastmilk [112,113]. Levels of DHA and EPA increase in breast milk within two to four days after initiation of supplementation with 10 mL per day of cod liver oil, while no change in the levels of ARA has been observed [111]. Helland et al. also reported that supplementation with cod liver oil increases DHA levels in a dose-response manner [111]. Therefore, nutritional strategies aiming at increasing DHA levels can be effective for replenishing DHA levels in women facing several pregnancies since DHA levels can drop over subsequent pregnancies [114]. Infants need high amounts of DHA for physiological development [115]. The breastmilk content in LC PUFAs is not regulated by the mammary gland but reflects the concentrations of LC PUFAs in maternal plasma that, in turn, are dependent on maternal diet and maternal activities of the desaturases and elongases involved in converting dietary LA and ALA to LC PUFAs [116].

FA from food sources in lactating mothers can be used in three ways: stored in adipose tissues, transferred to the mammary gland for incorporation into milk, or used for energy. However, a study showed that there is no effect of exercise on breastmilk content in LC PUFAs [117]. Different factors can influence the content of LC PUFAs of human breastmilk, such as maternal food intake, gestational age, or smoking. Infants of smoking mothers are present with fewer markers of LC PUFAs synthesis [118]. Concentrations of DHA and ARA in breastmilk presented variations depending on geographical locations, as reviewed earlier [104]. Breastmilk DHA content is linked to dietary intake [119]. Optimum breastmilk DHA concentrations were observed in artic Canada, Japan, the Dominican Republic, the Philippines, and the Congo (between 1.4% and 0.6%), which mainly contain coastal or insular populations, usually with high marine food intake [104]. Lower breastmilk DHA concentrations were observed in Pakistan, rural South Africa, Canada, the Netherlands, and France (between 0.06% and 0.14%), especially in inland regions or in developed countries, where low marine food consumption is observed [104]. A recent review reports that ARA is detected in animal and human milk, with its content being linked to maternal ARA dietary intake [120]. Interestingly, several recent works report the presence of free and esterified ARA, EPA, and DHA-derived oxylipins in human milk [121,122,123]. Of note, the most abundant forms of oxylipins are the ones derived from LA [122], which have been recently reported to be key to brain development in rats [124]. However, whether milk oxylipins play a role as signaling molecules for infant brain development has not been fully investigated.

PUFAs supply to premature infants is a burning question, as these fatty acids are massively incorporated in the developing brain in the last trimester of pregnancy. According to the World Health Organization, around 15 million premature births (before 37 weeks of gestation) occur every year. Prematurity is the main cause of death in children under the age of five [125]. Preterm infants have to face many neurodevelopmental disabilities, including learning, visual and hearing problems [126]. Current treatments consist of magnesium sulfate administration [127], caffeine for treatment of apnoea, and high doses of DHA [128]. To avoid complications in premature infants, different nutritional supports are used, such as enteral or parenteral nutrition, human breast milk, and formula milk [129]. Enteral nutrition is limited in preterm birth because of the immature gastrointestinal motor activity and risks of necrotizing enterocolitis [130]. Premature infants should be fed with human milk, while parenteral nutrition is recommended if *per os* nutrition cannot be achieved to supply adequate amounts of proteins. However, the best alternative to human milk remains preterm formula [131]. In 2008, a study conducted on very preterm infants with human milk supplementation using DHA and ARA brought interesting outcomes regarding improved recognition memory at six months of age [132]. However, the effect on visual, intellectual development, or growth of preterm infants with the addition of LC PUFAs to formula remains unclear [133,134]. Another study showed that supplementation with a 50:50 mixture of DHA:ARA had no negative effect on weight gain or growth [132]. A randomized control trial conducted in 2013 by Gould et al. underlined that preterm infants fed high-dose DHA did not display increased mental development index [135]. This can be explained by the fact that supplementation was conducted on fully- or partially-fed formula infants with fatty acids doses equivalent to the content found in human milk [136]. Assessments of attention had also been conducted by the team of Gould et al. on children from the n-3 Fatty Acids for Improvement in Respiratory Outcomes (N3RO) trial. The authors hypothesized that supplementation in DHA could favor the restoration of normal brain development [137]. Most of the studies carried out do not allow for a clear conclusion on the effect of LC PUFAs supplementation on cognitive development of preterm birth children [132,134,138,139], as also reported in a meta-analysis [140]. A recent review also reports mixed effects of LC PUFAs on cognition in preterm children [141]. A randomized controlled trial conducted on term infants in 2000 revealed that supplementation of formula milk with DHA and AA at an early stage of life improved the Mental Development Index of the Bayley Scales at 18 months of age [142]. Nonetheless, it is important to underline that neurodevelopmental assessment should be performed with neuropsychological tests and procedures adapted to the age of children.

To conclude, maternal dietary supply is crucial for the pre- and post-natal developing brain as the fetus and newborn are not able to produce LC PUFAs. The pre-natal maternal supply is accessed via the placenta and the BBB, thanks to specific transporters. The post-natal supply of LC PUFAs with breastmilk and formulas allows the transfer of nutrients to the developing brain. In the specific case of premature birth, formulas remain the best alternatives to human milk, even though long-term beneficial effects are still unclear.

## 3. Endogenous Production of PUFAs in the Developing Brain

### 3.1. Expression of Key Enzymes in the Developing Brain

As previously mentioned, maternal PUFAs supply to the infant brain occurs during gestation and lactation, but the developing brain seems to be able to form LC n-6 and n-3 PUFAs from their respective precursors. For a long time, it was assumed that the liver was the main location of elongation and saturation to form LC PUFAs, eventually released in the developing brain, as shown by in vivo studies in rats [51,143]. In order to bypass the metabolism of the liver, intra-cranial administration of labeled LA and ALA has been performed in the post-natal rat (11 and 13 days *post-partum*). These results show that the post-natal developing brain is capable of forming LC PUFAs, ARA, and DHA, respectively [144,145]. Similarly, Sanders et al. have shown that the brains of 21-day-old fetal rats were also capable of forming LC PUFAs, in particular by the presence of D6D, which converts LA and ALA [146,147]. At the end of the 1990s, the human *fads1* [148] and the mammalian *fads2* [148,149] were cloned and characterized, while the *fads1* gene (coding for D5D enzyme) has been shown to be strongly expressed in the human fetal brain [150]. In addition, several members of the Elovl family (Elovl 1, 3, 4, 5, and 6) have been reported to be expressed in the brain, with strong differences between species. As an example, the expression of Elovl2 and Elovl7 is very low in the mammalian brain, while Elovl2 is expressed in non-mammal species (such as fish). Elovl4, which catalyses the synthesis of very LC PUFAs, is expressed in the brain of fish and mammals. Several SNPs in the *fads1-fads2* gene cluster (rs174537, rs174761, and rs383458) are associated with variations in desaturase activity, leading to different LC PUFAs levels, as summarised in Table 1. An increase in EPA and DHA dietary intake is responsible for an increase in D5D and a decrease in D6D activities [151]. In one study, the authors showed that the presence of the rs174537 genotype resulted in dietary EPA and DHA modulation of D5D activity [151]. Besides, a 2011 meta-analysis on genetic loci associated with plasma phospholipids showed that SNPs in *Fads1/2* are responsible for an increase in ALA levels and a decrease in EPA levels, while SNPs on *Elovl2* are more likely to induce an increase in EPA (as well as docosapentaenoic acid, DPA) and a decrease in DHA plasma levels [152]. Mutations in either *elovl4* or *elovl5* cause neurological diseases in humans [153], as explained below. During early embryogenesis in the zebrafish, Monroig et al. were able to detect expression of the desaturase and elongation genes, using qRT-PCR and whole-mount in situ hybridization, respectively showing temporal- and spatially-restricted expression [154]. They showed that the three transcripts *fads*, *elovl2,* and *elovl5* were highly expressed in the head area from the beginning of embryogenesis, which was also confirmed in another study [155]. Early detection of these genes in the brain during embryonic development may suggest that in situ production of LC PUFAs could be achieved; however, the contribution of these genes as compared to the maternal PUFA supply have been poorly studied during brain development.

### 3.2. Gender Differences in Brain PUFAs Accretion and Effect of PUFAs on Sex Determination

In humans, it is known that the brain develops differently depending on gender. In a large human cohort, significant gender differences were observed regarding cortical thickness, fiber organization, and total brain volume [156]. These anatomical differences could explain that gender may play a significant role in neurodevelopmental disorders, as observed with autism [157] and attention deficit and hyperactivity disorder [158]. Besides, gender differences are also observed in patients with schizophrenia [159]. However, these differences are sometimes attributed to methodological issues [160,161].

To our knowledge, there are very few studies on differential brain accretion of PUFAs during development according to gender. In addition, previous studies did not focus on whether the dietary status of PUFAs differentially influences brain accretion according to gender. Nevertheless, some studies reported that PUFAs accretion is different in men vs. women. Indeed, in humans, DHA status in serum is reported to be higher in adult women than men. These differences have been associated with a better conversion of ALA to LC n-3 PUFAs in women as compared to men [162,163]. Such metabolic capacity could facilitate maternal supply to their offspring. On the other hand, the proportion of DPA n-3 to EPA is lower in serum of women [162,164]. Studies in rats have shown that D5D and D6D enzymes at both the transcriptional and protein levels are more expressed in the liver of females than males, in contrast to the brain, where similar expressions were found in both genders [165,166]. Different studies suggested that higher serum DHA levels in women than men may be due to the presence of estrogens that positively regulate DHA synthesis from ALA [166,167]. Indeed, it has been shown that taking hormonal contraception induces a significant increase in the level of DHA in women [168]. A recent study in rats has shown that a diet containing low amounts of LA with an estrogen supply induced an increase in the hepatic expression of D6D, Elovl2, and Elovl5, thus inducing an increase in plasma levels of DHA [169]. An in vivo study in rats showed that dietary ALA intake induced a greater proportion of DHA in erythrocytes but lower in the pre-frontal cortex in females compared to males [170]. In this study, the authors showed that ovariectomy, together with a diet rich in ALA, induced a decrease in the amount of DHA in erythrocytes, as well as in the hippocampus, compared to control females fed with a diet rich in ALA. Similarly, another study in rats also showed that ovariectomy-induced an increase on hepatic *fads1* and *fads2* transcripts, but not in the brain, and a decrease in DHA levels in the brain [171]. A study on deficiency or supplementation in n-3 PUFA during the perinatal period and for 16 weeks after weaning in mice, shows that the changes in cerebellar FA were more pronounced in offspring females, with diet having a significant effect [172], due to the presence of estrogen (for a review, see [173]). All these results suggest that ovarian hormones up-regulate DHA content in erythrocytes and brain regions. However, a recent study examining the interaction effects between diet, sex, brain regions, and phospholipid pools in mice demonstrated that DHA concentration was gender independent, while ARA concentration was partially dependent on sex [174].

Gender may influence the preventive effects of n-3 PUFA supplemented diets on neurodevelopmental disorders in animal models. In fact, supplementation with n-3 PUFAs in pregnant spontaneously hypertensive rat dams (SHR) induced a reduction in hyperactivity and impulsivity in the male offspring, but with no effect, or even opposite effects, in the female offspring [175]. A recent study conducted in a two-hit model in mice showed the sex-specific preventive effects of LC n-3 PUFAs [176].

All these studies show that there is a sex effect on the endogenous formation of LC PUFAs. However, further studies are needed to understand the effect of gender, and therefore hormones, on the accretion of PUFAs in the brain.

Trivers and Willard were the first to state that reproductive conditions could affect the sex ratio in the offspring [177]. Numerous in vivo studies have shown an impact of the maternal diet during pregnancy on the offspring sex ratio. Indeed, a low-fat diet (reduced amounts of essential fatty acids, EFA) during pregnancy induces a reduction in the number of males in the litter of mice without changing the total number of females [178]. In opossums, diets rich in LC n-3 PUFAs during gestation induced a greater proportion of males than females in litters [179], whereas a diet rich in n-6 LC PUFAs during gestation induced a greater proportion of females than males, both in mice [180] and rats [181]. Diets rich in n-6 LC PUFAs induced an increase in the production of pro-inflammatory derivatives, in particular prostaglandins, such as PGE_2_, PGF_2α_ and its metabolite 13,14-dihydro-15-keto PGF_2α_ (PGFM) [182]. This in utero inflammation affects the ovarian cycle, hormone production, and sperm fertilizing ability [183]. Besides, in utero inflammation also affects vaginal pH, inducing more favorable conditions for fertilization with X sperm rather than Y [184], as well as the loss of male embryos [181,182]. Surprisingly, it has been shown that a diet rich in LC n-6 PUFAs induces a higher proportion of males, both in sheep [185] and cows [186]. Various studies carried out in cows have shown that supplementation with n-6 LC PUFAs induced a reduction in the level of progesterone and a delay in oocyte maturation by an increase in the number of dominant follicles [187,188]. In cattle, it has been shown in vitro that the oocytes that mature later are preferentially fertilized by Y than X sperm [189] and that n-6 LC PUFAs supplementation during oocyte maturation and fertilization induced a greater number of male embryos [186]. Additional studies are necessary in order to understand these differences in results, as well as to decipher the different mechanisms that exert PUFAs on hormonal secretions and fertilization. Therefore, it appears that females have higher n-3 PUFAs in the serum, especially DHA, than males, due to the presence of estrogen, which positively regulates its synthesis [167].

## 4. Mechanisms of Action of PUFAs on Neurodevelopment

### 4.1. Role of PUFAs in Synaptogenesis and Neuronal Development

As previously mentioned, DHA increases during perinatal development, while the ARA/DHA ratio decreases, which is linked with active periods of synaptogenesis and the establishment of structural connectivity [190,191]. The mechanisms through which LC PUFAs contribute to brain development are still poorly understood. Indeed, several developmental processes have been reported to be regulated by PUFAs from cell migration to proliferation, differentiation, neurogenesis, and myelinisation to synaptogenesis, which are the most well-described and will be the main focus of the mechanisms described in this section.

Briefly, both DHA and ARA have been reported to influence neural stem cells (NSC) proliferation and differentiation, and neurogenesis. Early-life n-3 PUFAs dietary deficiency induces a delay in migration of neuronal cells in the embryonic brain [192] as well as in new-borns and during post-natal life [193]. Recent data pinpoint that n-3 PUFA deficiency-induced neurodevelopmental defects are linked to an early gliogenic fate shift in NSCs, with ARA and DHA derivatives being key [194]. Indeed, neurogenic transition of NSCs involves the DHA metabolite epoxydocosaexapentaenoic acids (EpDPE), while the gliogenic transition of NSCs is driven by the ARA metabolite epoxyeicosatrienoic acid (EET) [194]. In line with these results, other ARA metabolites such as PGE2 have been reported to increase the proliferation of NSCs and to promote their differentiation into neuronal-lineage cells [195]. DHA facilitates the differentiation of astrocytes in vitro [196] through its binding to GPR120 and β_2_-AR [197]. Of note, dietary supply in LA promotes ARA-derived endocannabinoids which, in turn, promotes astrogliogenesis from NSCs, reinforcing the idea that ARA metabolites contribute to gliogenesis [198]. Further studies on the exact role of LC PUFAS and their metabolites on NSCs fate (neurogenesis or gliogenesis) should be performed to confirm the importance of LC n-3 and n-6 PUFA dietary supply during brain development.

Regarding the role of LC PUFAs on synaptogenesis, a few lipid metabolism enzymes have been identified at synaptic terminals, where they can locally modulate synaptic transmission [199,200,201]. Mature synapses, formed during brain growth, require high amounts of ARA and DHA incorporated into the expanding membrane surface [202]. During perinatal rodent brain development, DHA modulates membrane signaling and synaptogenesis [203] by accumulating in the neuronal growth cone [204,205] and mature synaptic membranes [56,206]. An in vivo model of *Xenopus laevis* embryos from adult female frogs fed with adequate or n-3 PUFA deficient diets and then switched to a fish-oil supplemented diet showed that maternal n-3 PUFA intake impacts the branching and the synaptic connectivity of neurons in the developing brain. This model also revealed that these changes are correlated with a decrease in brain-derived neurotrophic factor (BDNF) in the brain [207]. Moreover, PUFAs-depleted drosophila presented impairments of synaptic transmission at synapses from the visual system [208]. Several in vitro studies have reported that DHA promoted synapse formation [40,209,210,211]. This effect could be direct, through DHA effect on specific receptors such as retinoid X receptor (RXR) [212] or indirect through specific metabolites such as oxylipins or N-docosahexaenoylethanolamine (DHEA), which have been reported to regulate NSCs differentiation (see below), neurite outgrowth, synaptogenesis and neuroinflammation [7,213,214]. Indeed, an ex vivo study on embryonic neurons from E18 mouse hippocampus’ showed that DHEA, a DHA-derived endocannabinoid-like metabolite, was crucial to neuronal development, promotes neurites and synaptogenesis [213]. Synapses enhancement and formation induced by DHA could also be linked to its esterified form via its effect on physical membrane properties [215,216,217,218]. As a result, in vivo DHA supplementation specifically promotes neurite growth, synaptogenesis, and raises the levels of pre- and post-synaptic proteins involved in synaptic transmission and long-term potentiation (LTP) [40].

Overall, the mechanistic data brought by animal studies further support the need for LC PUFA for optimal brain development. Concerning post-natal periods, breast milk remains the gold standard, as it provides both ARA and DHA [120]. Concerning infant formulae, the amount of ARA and DHA to be added is still a matter of debate, as recently reviewed [219]. The recent European recommendation does not pinpoint the need for the addition of ARA in infant formulae while adding DHA is mandatory (reviewed in [219]). The lack of recommendation on ARA in formulae has been pinpointed by several experts as being a potential risk for improper brain development [220,221,222], as a diet poor in ARA causes its decreased bioavailability for the brain [223]. Further studies are therefore needed to better understand the mechanisms underlying ARA and DHA association on brain developmental processes [224].

### 4.2. Role of PUFAs in the Regulation of Microglia and Neuroinflammation during Brain Development

More recently, specific attention has been given to the role of LC PUFAs on microglia activity and neuroinflammation during brain development [55]. Microglia, the resident macrophage-like brain cells which represent 5% to 15% of the brain cells, are well known to be crucial in neuroinflammatory processes in response to injury or lesions, including through interactions with brain infiltrating immune cells [225,226,227]. n-6 and n-3 PUFAs regulate neuroinflammation either directly or through their metabolites such as oxylipins or endocannabinoids (for a recent review, see [7]), including during brain development [228]. Of note, SNPs of genes involved in PUFAs metabolism have been reported to influence inflammation and potentially neuroinflammation. Indeed, children of mothers with minor alleles of *fads* rs174556 have been found to have higher ex vivo-stimulated production of IL-10, IL-17, and IL-5 from peripheral blood mononuclear cells [21].

Although microglia are known to contribute to neuroinflammation, accumulating evidence has pinpointed their role in brain wiring during development [229]. Microglia are the first glial cells appearing in the embryonic brain. The developmental origins of microglia have been the subject of intense debate [230]. A mesodermal or monocytic origin of microglia has been first hypothesized until fate-mapping studies showed that microglia derived from erythro-myeloid progenitors from the yolk sac around embryonic day (ED) 7.5 [231]. This primitive hematopoiesis gives rise to pre-macrophages that colonize the whole brain from ED 9.5 and differentiate into microglia from ED 10.5 [232]. Microglia develop according to temporal stages: early, pre, and adult, with the final step being reached during the second post-natal week in mice [233]. Such an early-life origin of microglia confers to these cells a specific life-long history, with gender and perinatal events potentially influencing their feature later in life [229]. During development, microglia regulate many processes, including the phagocytosis of alive neuronal elements (including synapses), dying and dead cells, and support myelinization, neurogenesis, and axon fasciculation [234,235]. Importantly, inflammatory events during pregnancy or early-life influence the microglial developmental trajectory, which may have adverse effects on brain wiring, increasing the risk for neurodevelopmental disorders [236,237,238,239]. Indeed, the mechanisms underlying normal and abnormal microglia-mediated synaptic pruning and brain wiring are the subject of intense research [236].

Overall, microglia functioning is largely driven by their microenvironment, with neurons and other glial cells constantly sending signals to which microglia respond. The role of dietary PUFAs in the regulation of microglia developmental features has been poorly addressed, but recent data pinpointed that these cells are influenced by dietary PUFAs. Recently, we reported that maternal n-3 PUFA intake influences the offspring microglial lipid composition (Figure 2) and the oxylipin signature [55,59]. Importantly, at weaning, microglia display a unique fatty acid profile, with an enrichment of EPA, suggesting that these cells could be a source of EPA-derived oxylipins with anti-inflammatory activities [59]. Indeed, resolvin D1 (RvD1) and resolvin E1 (RvE1), which are pro-resolutive oxylipins derived from DHA and EPA, respectively, reduced pro-inflammatory cytokine expression triggered in microglia by LPS in vitro [240]. On the other hand, maternal dietary n-3 PUFA deficiency polarizes microglia toward a phagocytic phenotype, which leads to altered brain wiring and memory impairment in the offspring at weaning [59,241]. In line with the importance of n-3 PUFAs in the regulation of microglia-dependant neuroinflammation [55,241,242,243,244,245], a low dietary level of maternal n-3 PUFAs not only promotes a pro-inflammatory microglial profile [241] but also exacerbates inflammatory response in pregnant dams and the embryos after a prenatal LPS treatment [243]. This exaggerated embryonic brain inflammatory response contributes to later-life cognitive, emotional, and neurobiological impairments [243,246]. The mechanisms underlying the deleterious effect of both prenatal dietary n-3 PUFA deficiency and inflammatory stimulus on later-life behavior have started to be understood; for example, the exacerbated production of inflammatory factors in the embryonic brain could disturb brain wiring [247], the microglia phagocytic activity of spines, leading to excessive pruning [248] and/or microbiota disturbance [246], which has been reported to be involved in neurodevelopmental disorders [249,250]. In particular, we recently found that 12-HETE, which is overexpressed in n-3 PUFA deficient microglia, contributes to dendritic spines decrease and memory impairment at weaning [59]. Additional work needs to be conducted to decipher the exact mechanisms underlying n-3 PUFA deficiency and developmental microglia function and whether later-life dietary intervention with n-3 PUFAs can restore the impairment associated with this early-life deleterious developmental effect [251].

To conclude, co-occurring n-3 PUFA deficiency and maternal inflammation can potentiate each other and induce synergistic effects on brain development and a higher risk of developing neurodevelopmental disorders (Figure 2). As a result, women in the age of procreating with low n-3 PUFA bioavailability (linked to diet and/or genetic) could be at risk of higher sensitivity to early-life adverse inflammatory events. Indeed, there is a need to pay attention to n-3 PUFAs bioavailability during pregnancy to limit the risk of impaired brain development and neurodevelopmental disorders.

## 5. Role of PUFAs in Neurodevelopmental Diseases

Neurodevelopmental disorders, which affect more than 3% of children worldwide, are characterized by altered or disrupted developmental steps leading to the incapacity to reach cognitive, emotional, communicative, and motor abilities, together with poor adaptation skills (recently reviewed in [252]). Among neurodevelopmental disorders, autism spectrum disorder (ASD), attention deficit hyperactivity disorder (ADHD), and intellectual disability are common, with potential shared neuropathological mechanisms. The factors contributing to neurodevelopmental disorders can be genetic and/or environmental. As an example, clinical and pre-clinical studies show that genetic alterations of genes coding for proteins involved in pre- and post-synapse functions have been found to be implicated in neurodevelopmental disorders. Schizophrenia, despite not being classified as a neurodevelopmental disorder, is a neuropsychiatric disease with an etiology route of inadequate brain development, with long-lasting consequences. Altogether, these diseases refer to the theory of DOHAD (Developmental Origin of Health and Disease; [253]), with early-life nutritional imbalance playing a key role [254].

Both genetic alteration and inadequate nutrition (amongst others), the latter leading to insufficiency of LC n-3 PUFAs bioavailability, have been reported to be risk factors for neurodevelopmental diseases [228] and schizophrenia [255]. Besides, abnormal features of the placenta (physiological, morphological, and histological abnormalities) have been hypothesized as risk factors for subsequent abnormal neurodevelopment [256,257]. In this section, the potential role of LC PUFAs in neurodevelopmental diseases is discussed, especially the role of the n-3 family. In particular, we focus on the link between perinatal low LC n-3 PUFAs and the risk of compromised brain development and whether dietary supplementation with LC PUFAs could protect and/or correct neurodevelopmental abnormalities.

### 5.1. Early-Life n-3 PUFAs and Cognition in Infants

In humans, several studies have reported a link between n-3 PUFAs and cognitive capacities. Explanations have been focused on the balance (ratio) between DHA and ARA, as emphasized before [258,259,260]. Indeed, an association between low levels of DHA and poorer reading abilities and working memory performance was observed in children [261]. Besides, a negative relationship was found between EPA-DHA consumption and overall cognitive function and psychomotor speed in subjects aged 45–70 years old [262]. Moreover, beneficial effects of maternal n-3 PUFA supplementation on child growth and development have been reported [263], suggesting a positive effect on cognitive performance in later life. Due to the observed positive association between n-3 PUFA serum levels and cognitive abilities in infants [261,264], there have been various attempts to enhance cognition through LC n-3 PUFAs supplementation in healthy children and adolescents [71,265,266,267,268] as well as in youths from clinical populations, such as patients with autism spectrum disorder (ASD) or attention deficits and hyperactivity disorder (ADHD) [269,270], which will be discussed below.

Different studies have shown that n-3 PUFAs dietary supplementation of the mother or the infant could have beneficial effects on development and cognition in children. In fact, the intelligent quotient (IQ) can be improved at four years of age after LC n-3 PUFA supplementation (1183 mg of DHA and 803 mg of EPA per day) in the mother during pregnancy and lactation [271]. Supplementation of infant formula with DHA (0.36%) and ARA (0.72%) from an early age (four to six months) leads to improved visual function at 12 months [272]. Using the same paradigm, another study found improved visual acuity at six weeks of age [273]. Other studies have also shown that there is a positive correlation between formula rich in ALA and neurodevelopment at one year of age in infants born at full term of pregnancy [274]. It has also been shown that dietary LC PUFAs supplementation (using both DHA and ARA) can improve visual acuity and cognitive development [275]. However, the duration of breastfeeding, without dietary supplementation and the performance score of infants are weakly associated [276]. These studies strongly suggest that supplementation with both DHA and ARA during early life can provide beneficial effects on cognition in infants. However, future studies will need to decipher whether these effects are provided by either n-3 or n-6 LC PUFAs, or the synergy of both.

In parallel, several animal studies have examined the impact of LC PUFAs on cognition. In a model of transgenic mice producing high levels of LC n-3 PUFAs in their milk under dietary n-3 PUFAs deficiency (beta-casein n-3 desaturase transgenic mouse model), pups raised under these conditions had increased brain levels of DHA and faster visual development compared to pups raised by wild type mice [277]. In a review published in 2006, Fedorova and Salem provided an overview of the different animal models used to investigate the cognitive and behavioral effects of LC n-3 PUFAs [278]. Most of these models use LC n-3 PUFAs dietary deficiency; however, it is important to note that this deficiency is far more severe than what can be found in human populations [136]. In mice, n-3 PUFAs deficient diets during pregnancy and lactation induce spatial memory deficits and behavioral impairment in the adult offspring [243,244,279,280].

### 5.2. Autism Spectrum Disorders (ASD)

Autism Spectrum Disorders (ASD) are neurodevelopmental diseases more frequently diagnosed in males [281]. ASD is characterized by an early-childhood onset (first years of life) with a long-term course and variation in developmental trajectory, a remarkable clinical and biological variability, and a dramatically increased prevalence (0.05% in 1966 and close to 2% in 2019), with a wide symptomatology range in patients [282]. Patients with ASD display a core of symptoms such as impairment in social-communicative skills and restricted/repetitive behaviors/interests, often associated with other symptoms such as intellectual disabilities, impaired language skills, and different medical conditions such as gastrointestinal symptoms.

The substantial heterogeneity which underlies the neurobiology of ASD further supports that the etiology and pathophysiology of these disorders cannot be restricted to a single genetic cause [283]. Indeed, genetic alterations [284], pollution, environmental/gestational stress, including inflammation and nutrition [285,286], as well as neural/anatomical dysfunctions [287,288,289,290] are combined risk factors for ASD development. With regard to the link between ASD and PUFAs, polymorphisms in either the *fads* or *elovl* genes have been linked to susceptibility for developing ASD [291].

In rodents, exposure to an n-6-PUFAs-rich diet during gestation and lactation produced social deficits in the adult offspring that resemble autistic features [292]. In a rat model of autism, where pups are exposed to the neurotoxin propionic acid, decreased levels of total n-3 and n-6 PUFAs were observed in the brain [293]. Another study found beneficial effects of LC n-3 PUFAs (200 mg/kg/day for 30 days) in the same model [294]. In another study, in which mice pups were exposed to an immune reaction during intrauterine life (intraperitoneal injection of lipopolysaccharide at E17 in pregnant mice, mimicking bacterial infection), adult animals developed impaired memory if fed with an LC n-3 PUFAs deficient diet [243]. In another model of intrauterine exposure to infection (exposure to PolyI:C on E9, mimicking viral infection), supplementation with LC n-3 PUFAs at weaning (menhaden fish oil at 35 g/kg of diet) dampened the DNA hypo-methylations observed at adulthood [295]. In a mouse model of ASD, in which animals are prenatally exposed to valproic acid, n-3 supplementation with both α- and γ-linolenic acid protected against the development of autistic-like features [296]. Besides, in an inbred mouse model of ASD (BTBR mouse strain), dietary deficiency in n-3 PUFAs from gestation to early adulthood induced developmental delay and altered sociability [297]. These results were also observed when these animals were fed with dietary n-3 PUFAs supplementation [297], suggesting that LC n-3 PUFAs cannot counterbalance the social deficits induced by such a genetic inbreeding. Finally, in a study of Frm1 KO mice, another model mimicking ASD-like features, n-3 PUFAs supplementation from weaning to adulthood, led to significant improvements in sociability and emotionality [298], suggesting that n-3 PUFAs supplementation might be used as a therapeutic tool in specific clinical situations.

PUFAs have been investigated for their potential role in alleviating the symptoms of autism since reduced levels of PUFAs have been observed in autistic patients, especially concerning ARA and DHA [299,300,301,302,303,304,305], which are believed to be correlated to the symptomatology of ASD [306]. However, one study observed increased levels of PUFAs in high-functioning autistic children [307]. Besides, a decreased risk of ASD in children born from mothers with high total dietary PUFA intake was also observed [308]. Another study found that low dietary PUFAs intake during the second half of pregnancy appears to be a risk factor for ASD [309]. In an open-label study, a six-week supplementation with fish-oil capsules containing EPA + DHA (1.86 g/day) and vitamin E (10 mg/day) failed to improve the behavioral symptoms of young adults with severe autism [310]. In a systematic review, PUFAs supplementation with either DHA or EPA alone, or in combination, was found to be inefficient in modulating behavioral outcomes in adults and adolescents with autism [311]. However, a recent case report provided evidence that supplementation with EPA + DHA (respectively 2.4 and 1.2 g/day) and vitamin D (25,000 IU per week in a single oral dose) are beneficial to improve the symptoms of a 23 years old autistic patient [312]. Similarly, a meta-analysis performed recently concluded that n-3 PUFAs supplementation could be effective in improving some of the symptoms of autistic patients [305]. A recent randomized clinical trial observed beneficial effects of n-3 PUFAs supplementation (722 mg/day of DHA for 12 months) on irritability and lethargy in 2.5–8 years old autistic children [313], which was also observed in meta-analyses [314]. The beneficial effects of LC PUFAs were also observed in preterm toddlers presenting with ASD symptoms [315,316]. Finally, supplementation with both LA (480 mg/day) and ALA (240 mg/day) for 16 weeks provided therapeutic benefits in 21 patients with ASD [304]. Table 2 summarises the main outcomes of case reports [312,317], open-label clinical trials [304,310,318,319,320], and randomized clinical trials [313,315,316,321,322,323,324,325,326,327,328,329] using LC PUFAs in patients diagnosed with ASD. An elegant study has summarised the wide array of possible nutritional interventions for patients with ASD and concluded that dietary interventions have little effect on ASD symptomatology [330]. Recently, a meta-analysis conducted on four randomized clinical trials revealed significant improvements in the symptoms presented by patients with ASD [305].

To conclude, the effects of LC n-3 PUFAs in clinical trials on ASD have led to mitigating effects. As different risk factors can lead to ASD development, it is quite unclear which treatments or dietary interventions could be efficient in preventing symptoms. However, some clinical studies highlight the beneficial effect of dietary supplementation on symptoms in young autistic patients.

### 5.3. Attention Deficits and Hyperactivity Disorder (ADHD)

Attention deficits and hyperactivity disorder (ADHD) is classified as a neurodevelopmental disorder. The prevalence of ADHD is around 8% [331] and is diagnosed in children. Some patients may also develop ADHD into adulthood [332]. According to the Diagnostic and Statistical Manual of Mental Disorders (DSM, fifth version), symptoms of ADHD include impulsivity, inattention as well as social and academic difficulties, although a certain fluidity exists [333]. Depending upon the symptoms, three different types of ADHD can be distinguished: predominantly difficulty in concentration, predominantly hyperactivity and impulsiveness, and finally, a combination of all of the above [334]. ADHD can include a wide range of symptoms such as restlessness, fidgeting, anxiety, attention deficit, distractibility, excessive talking, forgetfulness, and frequent interruption of others [335]. In fact, in several studies, low birth weight, premature birth, infections, or traumas could be considered as potential causes of ADHD [336,337]. A few studies have also found an association between several protein-coding genes and ADHD. These include dopamine transporters (DAT_1_), dopamine (D_4_, D_5_), serotonin (5-HT_1B_), and NMDA 2A receptors [338,339,340]. Finally, morphological abnormalities were found in the brain of ADHD patients [341,342,343], although this is frequently disputed in the field. The etiology of ADHD needs to be clarified more precisely.

Patients with ADHD present lower levels of PUFAs in the blood [344,345,346,347,348], which appear to correlate to the symptomatology of ADHD [349]. In a recent cohort of children, lower dietary intakes of fatty fish and seafood were observed in ADHD patients compared to control patients [350]. Thus, strategies aiming at increasing serum levels of PUFAs have been investigated as potential dietary treatments for the management of ADHD symptoms.

Similar to ASD models, ADHD models in rodents are numerous [351]. Since dopaminergic neurotransmission is markedly altered in ADHD [352,353,354,355,356], animal models have focused on genetic alterations of dopaminergic neurotransmission. Indeed, some studies used dopamine D_2_ autoreceptor KO mice to investigate ADHD-like symptoms. These mice display spontaneous hyperlocomotion [357] and impulsivity [358], two features frequently observed in ADHD [359]. Other studies used dopamine transporter (DAT) KO rodents, which also present similarities with human symptomatology. In fact, DAT-KO animals present traits of hyperactivity and motor stereotypy [360,361,362,363,364]. Some studies have evaluated the potential effects of PUFAs on spontaneous hypertensive (SHR) rats, a rodent model of ADHD [365]. Indeed, a study showed a significant correlation between a low level of n-3 PUFAs in the prefrontal cortex in rats and locomotor hyperactivity [366]. Interestingly, in the SHR rat, a reduction in hyper-locomotion was observed following dietary enrichment with LC n-3 PUFAs (LA 1.54% and ALA 0.27%) compared to deficiency in LC n-3 PUFAs (LA 1.58% and ALA 0.01%) [367]. An EPA and DHA-enriched diet during pregnancy in SHR dams enhanced reinforcement-controlled attention in males, and reduced hyperactivity and impulsiveness, while there is no change in females [175]. The authors explained this sex-specific effect to be due to increased turnover ratios of dopamine and serotonin for SHR males in the neostriatum, while there was no change for the females SHR.

A double-blind, randomized clinical trial performed in Iran revealed subtle improvements in ADHD symptoms in patients under standard pharmacology (methylphenidate 1 mg/kg/day) combined with n-3 PUFAs supplementation (EPA 180 mg/day and DHA 120 mg/day) after eight weeks, compared to controls (pharmacological treatment alone) patients [368]. Similarly, n-3 PUFAs supplementation with 93 mg/day of EPA and 29 mg/day of DHA during 15 weeks was able to alleviate ADHD-like symptoms in 104 children [369]. Identical results were found by another study, in which 20–25 mg/kg/day of EPA and 8.5–10.5 mg/kg/day of DHA were consumed for 16 weeks [370], while earlier studies revealed mild improvements in such treatment [371,372]. A recent study on ADHD children (6–18 years old) observed improvements in attention and vigilance after 12 weeks of EPA supplementation (1.2 g/day), while attention was worsened by the treatment in patients presenting higher endogenous EPA levels at enrolment [270]. However, one clinical trial in ADHD children (6–12 years old) found no beneficial effects of PUFAs supplementation (241 mg/day of DHA, 33 mg/day of EPA, and 150 mg/day of n-6 PUFAs) after 10 weeks of treatment [373], when combined with the standard pharmacological treatment (methylphenidate 1 mg/kg/day). A meta-analysis performed on 699 children with ADHD from 10 clinical trials revealed a small but significant effect of n-3 PUFAs supplementation on ADHD symptoms [269], while another study failed to find beneficial effects of such a supplementation [374], which might be explained by very different methodologies in all of the clinical trials investigated. Finally, it was recently suggested that the combination of EPA and DHA could be used as an adjuvant therapy, together with the standard pharmacological treatment, using methylphenidate, to improve the clinical symptomatology in ADHD patients [375]. Table 3 summarises the main outcomes of an open-label trial [375] and randomized clinical trials [270,368,369,370,371,372,373,376,377,378,379,380,381,382,383,384,385,386,387,388,389,390,391,392,393,394,395,396,397,398] using LC PUFAs in patients with ADHD.

To conclude, the effects of LC n-3 PUFAs supplementation on ADHD patients are ambiguous. While some studies report beneficial effects, others report no significant effects.

Interventions using LC n-3 PUFAs in patients with neurodevelopmental disorders produced mitigated results. Indeed, in patients with ASD, 11 out of 18 interventions produced beneficial effects (Table 2), while 21 out of 31 interventions produced beneficial/minimal effects in patients with ADHD (Table 3).

### 5.4. Schizophrenia

Schizophrenia is a complex disorder, which affects 1% of the population worldwide. The main symptoms are hallucinations, delusions, disorganized thought, blunted or inappropriate affect, social withdrawal, and cognitive dysfunction [399]. However, the etiology is still unclear.

The possibility that schizophrenia might arise from abnormal neurodevelopment was hypothesized long ago [400,401,402,403]. Emerging evidence suggests that a combined neurodevelopment insult, together with another insult, such as inflammation [404], infection [405], psychological trauma [406], consumption of cannabis [407], or malnutrition [408,409], known as the two-hit hypothesis, can lead to schizophrenia. Dysfunction of a variety of neurotransmitter systems (glutamatergic, serotonergic, and GABAergic) has been reported in schizophrenia [410], while the brain dopamine systems seem to play an important role in the disease [411]. Indeed, hyperactivity of the mesolimbic dopamine pathway could mediate the psychotic symptoms of schizophrenia. In addition, hypofunction of the prefrontal cortex, which appears to involve decreased activity of mesocortical dopamine neurons, has been reported in schizophrenic patients [412]. Importantly, effective anti-psychotic drugs are dopamine D2 receptor (D2R) antagonists. A number of clinical studies report a contribution of LC n-3 PUFA to schizophrenia. In fact, psychosis itself is associated with an imbalanced dietary intake of LC PUFAs [413,414]. Not only low levels of PUFAs have been associated with schizophrenia, but also some genetic alterations of genes involved in the transport and the metabolization of PUFAs have been reported in schizophrenic patients. As an example, SNPs in *Fabp7*, a transporter of LC PUFAs, and *ALOX12*, a gene encoding an enzyme, which metabolizes both ARA and DHA into oxylipins, have been linked to schizophrenia [415,416]. FADS2 and iPLA2, which hydrolyzes DHA in the cell membrane, are highly expressed in the brain of schizophrenic patients [417,418]. Importantly, it is well known that genetic or nutritional suboptimal levels of n-3 PUFA influence the dopaminergic system [419], which is highly involved in several neuropsychiatric disorders, including schizophrenia (recently reviewed in [255,420].

In rodents, prenatal deficiency in LC PUFAs (ARA and DHA) could model the prodromal state of schizophrenia [421]. Dietary PUFA deficiency resulted in schizophrenic-like symptoms, dysregulated expression of genes involved in oligodendrocyte and GABA activity, and epigenetic silencing of the lipid receptors Rxr and Ppar, which have also been reported in patients [421]. Similarly, LC n-3 PUFAs deficiency during gestation resulted in prepulse inhibition impairments in mice [422], which parallels the abnormal prepulse inhibition responses in patients with schizophrenia [423,424,425]. Interestingly, *Fabp3* KO mice, which have impaired brain PUFAs access, exhibit D2R dysfunction, impaired glutamate release in the dorsal striatum and the anterior cingulate cortex [426,427] (ACC), which could be hallmarks of schizophrenia and ADHD [428,429]. A causal link between early-life dietary n-3 PUFA deficiency and dopaminergic system impairment has recently been demonstrated in mice, further reinforcing the importance of dietary n-3 PUFA in dopaminergic function [430]. This reinforces the demonstration that striatal D2R neurons are particularly sensitive to fatty acids in rodents [431]. Indeed, both clinical and preclinical studies suggest that inadequate supply of LC n-3 PUFAs during early brain development leads to altered dopaminergic function and schizophrenic symptoms. This corroborates the observation of reduced DHA in the blood and brain of schizophrenic patients. Altogether, these data support the need for adequate n-3 PUFA supply during the perinatal period to promote adequate neurodevelopment, in particular of the dopaminergic system.

As in other neurodevelopmental diseases, the question of correcting schizophrenic symptoms through the use of dietary supply in LC n-3 PUFAs has been questioned in several clinical studies leading to puzzling results. One study observed preventive effects of LC PUFAs (mainly EPA 700 mg/day, DHA 480 mg/day, and vitamin E 7.6 mg/day, during 12 weeks) on the development of psychosis [432], probably via mechanisms implicated in myelination [433,434]. A recent systematic review has summarized all the interventional studies focused on the link between psychosis and LC PUFAs [435]. Patients suffering from schizophrenia present decreased levels of LC PUFAs in the brain [436] and erythrocytes [437,438,439]. As such, a study investigated the potential therapeutic benefits of LC PUFAs in schizophrenia. Sixteen weeks of supplementation with LC PUFAs (EPA 700 mg/day and DHA 400 mg/day) provided therapeutic improvements in patients [440]. A 2016 review has summarized all previous clinical trials using LC n-3 PUFAs on schizophrenia [441]. While some clinical trials have proven clear beneficial effects of LC n-3 PUFAs, others resulted in no significant effects [441].

## 6. Conclusions and Future Directions

The understanding of the mechanisms underlying perinatal dietary PUFAs to physiological and pathological brain development remains incomplete. However, significant advancements have been made in recent years, with more understanding of how low levels of LC PUFAs (nutritional and/or genetic) during pregnancy and infancy contribute to the etiopathology of neurodevelopmental disorders. This knowledge is essential to design appropriate nutritional intervention with LC PUFAs in mothers and children with low levels of LC PUFAs, children at risk of impaired neurodevelopment, or women facing at-risk pregnancies. A more comprehensive understanding of the genetic, physiological, and behavioral modulators of EPA and DHA status and response to intervention is needed to allow refinement of current dietary LC n-3 PUFA recommendations and stratification of advice to “vulnerable” and responsive subgroups. Overall, PUFA-based gene-diet interactions in humans should provide a solid scientific basis for the development of personalized nutritional intervention early in life.

## Figures and Tables

**Figure 1 nutrients-13-01185-f001:**
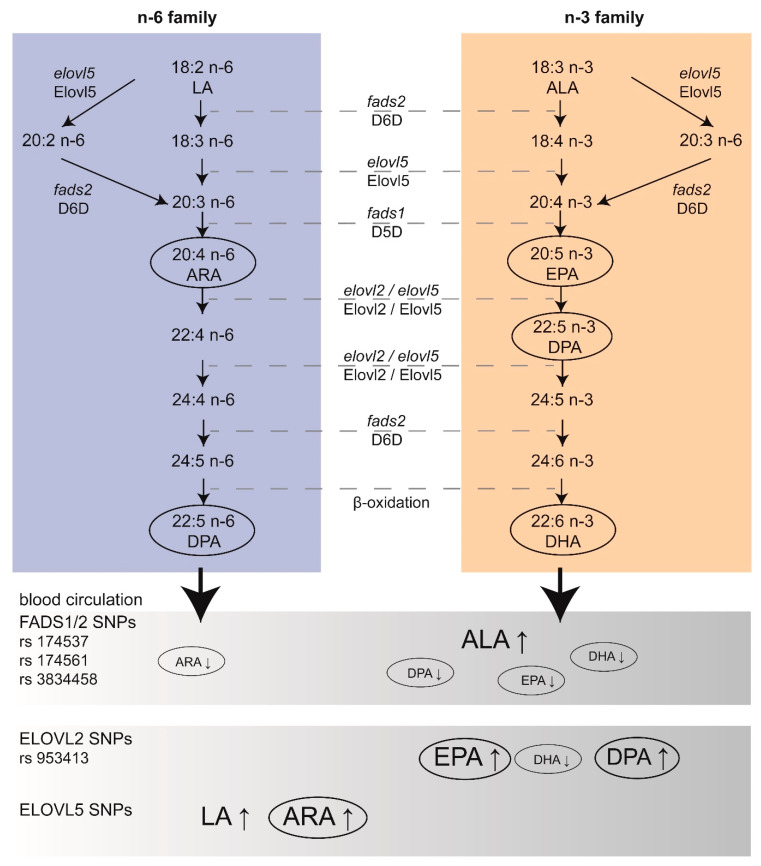
Single nucleotide polymorphisms (SNPs) within the metabolic pathways of n-3 and n-6 PUFAs lead to altered circulating levels of PUFAs. Desaturase enzymes (delta-5 desaturase (D5D) and delta-6 desaturase (D6D)) encoded by fatty acid desaturase genes 1 and 2 (fads1 and fads2 respectively), as well as elongase enzymes (elongase 2 (Elovl2) and elongase 5 (Elovl5)) encoded by elovl2 and elovl5 genes (respectively), are involved in desaturation and elongation of PUFAs from the n-6 (cyan) and n-3 (orange) families. Polymorphisms in those genes (SNPs) are associated with variations of plasma (grey) levels of LC PUFAs (represented by circles in the blood circulation). Bold letters represent higher amounts of LC PUFAs in the blood. fads: fatty acid desaturase; elovl: elongase; EPA: eicosapentaenoic acid; ARA: arachidonic acid; DHA: docosahexaenoic acid; ALA: α-linolenic acid; LA: linoleic acid; DPA: docosapentaenoic acid; D5D: delta-5 desaturase; D6D: delta-6 desaturase; Elovl5: elongase 5; Elovl6: elongase 6.

**Figure 2 nutrients-13-01185-f002:**
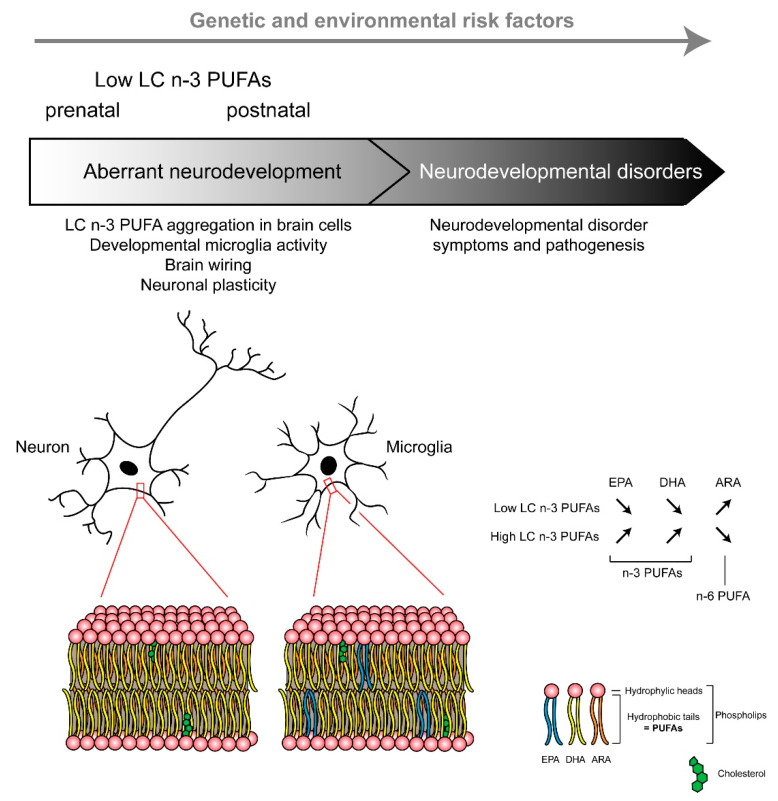
Hypothetical summary of events leading to neurodevelopmental disorders. Aberrant neurodevelopment can lead to neurodevelopmental disorders following inadequate levels of LC n-3 PUFAs.

**Table 1 nutrients-13-01185-t001:** Major SNPs involved in variations of LC PUFAs levels. *fads:* fatty acid desaturase; *elovl:* elongase; LC n-3 PUFA: n-3 long chain polyunsaturated fatty acid; EPA: eicosapentaenoic acid; ARA: arachidonic acid; DHA: docosahexaenoic acid; ALA: α-linolenic acid; LA: linoleic acid; DPA: docosapentaenoic acid; D5D: delta-5 desaturase; D6D: delta-6 desaturase.

Study (Year)	Ref.	SNPs	Impact on LC PUFAs Levels	Child Development
Morales (2011)	[18]	Mother: *fads* *elovl5* Children: *fads* *elovl5*	Higher colostrum levels of LC n-3 PUFA.	Improved cognition at 14 months. Modification of cognition by breastfeeding.
de Groot et al. (2019)	[19]	*fads1-fads2* (rs174537, rs175461, rs3834458) *elovl2* (rs953413)	Decreased levels of EPA and ARA. Reduced DHA levels.	
Tanaka et al. (2009)	[20]	*fads* (rs174537) *elovl2* (rs953413)	Higher ALA, LA and lower ARA, EPA, DPA and DHA levels. Lower DHA and higher DPA and ARA levels.	
Al-Hilal et al. (2013)	[151]	*fads1-fads2* (rs174537)	Interaction of rs174537 genotype with dietary EPA + DHA supplementation influence D5D and D6D activity.	
Lemaitre et al. (2011)	[152]	*fads1-fads2*	Increase in ALA levels. Decrease in EPA levels.	
		*elovl2*	Increase in EPA/DPA levels. Decrease in DHA levels.	

**Table 2 nutrients-13-01185-t002:** Interventional studies using LC PUFAs in patients with autistic spectrum disorder (ASD) or ASD-like symptoms. ALA: α-linolenic acid; ARA: arachidonic acid; DHA: docosahexaenoic acid; EPA: eicosapentaenoic acid; GLA: γ-linolenic acid; LA: linoleic acid, n-9: n-9 PUFAs.

Design	Study (Year)	Ref.	Final Sample Size	Duration	Intervention	Main Outcome(s)
Case reports	Johnson (2003)	[317]	One patient	4 weeks	EPA 540 mg/day	Significant improvements in symptoms.
	Infante (2018)	[312]	One patient	24 months	EPA 204 g/day, DHA 1.2 g/day, vitamin D 25,000 IU/week	Beneficial improvements in psychiatric symptoms.
Open-label trials	Politi (2008)	[310]	19 interventions	6 weeks	EPA + DHA 1.86 g/day, vitamin E 10 mg/day	No significant effects.
	Meiri (2009)	[318]	Nine interventions	12 weeks	EPA 380 mg/day, DHA 180 mg/day	Some improvements.
	Johnson (2010)	[320]	10 interventions	3 months	DHA 400 mg/day	No significant effects.
	Ooi (2015)	[319]	41 interventions	12 weeks	DHA 840 mg/day, EPA 192 mg/day, ARA 66 mg/day, GLA 144 mg/day, vitamin E 60 mg/day, thyme oil mg/day	Improved core symptoms.
	Yui (2016)	[304]	21 controls and 21 interventions	16 weeks	LA 480 mg/day, ALA 240 mg/kg	Beneficial effects on aberrant behaviours and social responsiveness.
Randomised clinical trials	Amminger (2007)	[321]	Five controls and seven interventions	6 weeks	EPA 0.84 g/day, DHA 0.7 g/day	Improved hyperactivity and stereotypy.
	Bent (2011)	[322]	12 controls and 13 interventions	12 weeks	EPA 0.7 g/day, DHA 0.46 g/day	No significant effects.
	Yui (2011 and 2012)	[323,324]	Six controls and seven interventions	16 weeks	DHA 0.24 g/day, ARA 0.24 g/day	Improvements in social withdrawal and stereotypy.
	Bent (2014)	[325]	28 controls and 29 interventions	6 weeks	EPA 0.7 g/day, DHA 0.46 g/day	No significant effects.
	Voigt (2014)	[326]	15 controls and 19 interventions	6 months	DHA 0.2 g/day	No significant effects.
	Mankad (2015)	[327]	19 controls and 18 interventions	6 months	EPA + DHA 1.5 g/day	No significant effects.
	Parellada (2017)	[328]	68 patients	8 weeks	EPA 577.5–693 mg/day, DHA 385–462 mg/day, vitamin E 1.6–2.01 mg/day	Within subject improvements in social motivation and social communication.
	Boone (2017)	[329]	16 controls and 15 interventions	3 months	EPA 338 mg/day, DHA 225 mg/day, GLA 83 mg/day, n-9 306 mg/day	No significant effects.
	Sheppard (2017)	[315]	12 controls and 12 interventions	3 months	EPA 338 mg/day, DHA 225 mg/day, GLA 83 mg/day, n-9 306 mg/day	Improved gesture and word use.
	Keim (2018)	[316]	16 controls and 15 interventions	3 months	EPA 338 mg/day, DHA 225 mg/day, GLA 83 mg/day, n-9 306 mg/day	Improvements in symptoms, but limited to one subscale.
	Mazahery (2019)	[313]	16 controls and 23 interventions	12 months	DHA 722 mg/day, vitamin D 2000 IU/day	Improved irritability.

**Table 3 nutrients-13-01185-t003:** Interventional studies using LC PUFAs in patients with ADHD. ALA: α-linolenic acid; ARA: arachidonic acid; DHA: docosahexaenoic acid; EPA: eicosapentaenoic acid; GLA: γ-linolenic acid; LA: linoleic acid; n-3: n-3 PUFAs; n-6: n-6 PUFAs.

Design	Study (Year)	Ref.	Final Sample Size	Duration	Intervention	Main Outcome(s)
Open-label trial	Checa-Ros (2019)	[375]	40 patients	1 month	EPA 70 mg/day, DHA 250 mg/day, methylphenidate 1 mg/kg/day	Improved attention, improved core symptoms
Randomized clinical trials	Aman (1987)	[376]	31 patients	4 weeks	LA 2.16 g/day, GLA 270 mg/day	Only minimal effects.
	Voigt (2001)	[377]	27 controls and 27 interventions	4 months	DHA 345 mg/day	No significant effect.
	Richardson (2002)	[371]	14 controls and 15 interventions	12 weeks	EPA 186 mg/day, DHA 450 mg/day, ALA 96 mg/day, LA 864 mg/day, ARA 42 mg/day, vitamin E 60 IU/day, thyme oil 8 mg/day	General behavioral improvements and small effects.
	Stevens (2003)	[372]	22 controls and 25 interventions	4 months	EPA 80 mg/day, DHA 480 mg/day, GLA 96 mg/day, ARA 40 mg/day, vitamin E 24 mg/day	No clear benefit but some improvements.
	Hirayama (2004)	[378]	20 controls, 20 interventions	2 months	DHA 3.6 g/week	Only minimal effects.
	Sinn (2007)	[369]	27 controls and 36 interventions	15 weeks	EPA 558 mg/day, DHA 174 mg/day, GLA 60 mg/day, vitamin E 10.8 m/day	Improvements in core symptoms.
	Sinn (2008)	[379]	27–38 controls and 72–88 interventions	15–30 weeks	EPA 558 mg/day, DHA 174 mg/day, GLA 60 mg/day	Improved attention.
	Vaisman (2008)	[380]	21 controls and 21 interventions	3 months	Fish oil 799 mg/day	Improved attention.
	Bélanger (2009)	[370]	13 controls and 13 interventions	16 weeks	EPA 20–25 mg/kg/day, DHA 8.5–10.5 mg/kg/day, vitamin E in traces	Improvements in core symptoms.
	Johnson (2009)	[381]	38 controls and 37 interventions	3–6 months	EPA 558 mg/day, DHA 174 mg/day, GLA 60 mg/day, vitamin E 10.8 mg/day	Improved symptoms in a subgroup.
	Raz (2009)	[382]	31 controls and 32 interventions	7 weeks	LA 480 mg/day, ALA 120 mg/day, vitamin E 10 mg/day, mineral oil 190 mg/day	No significant effect.
	Gustafsson (2010)	[383]	42 controls and 40 interventions	15 weeks	EPA 500 mg/day, DHA 2.7 mg/day, vitamin E 10 mg/day	Improved attention.
	Hariri (2012)	[384]	50 controls and 53 interventions	8 weeks	EPA 635 mg/day, DHA 195 mg/day	Improved symptoms.
	Manor (2012)	[385]	47 controls and 100 interventions	15–30 weeks	EPA 80 mg/day, DHA 40 mg/day	Improved impulsivity.
	Milte (2012)	[386]	90 patients	12 months	EPA 264–1109 mg/day, DHA 108–1032 mg/day, LA 1467 mg/day	Improved symptoms in a subgroup.
	Milte (2015)	[387]	87 patients	12 months	EPA 264–1109 mg/day, DHA 108–1032 mg/day, LA 1467 mg/day	Improved symptoms in a subgroup.
	Perera (2012)	[388]	46 controls and 48 interventions	6 months	n-3 592.7 m/day, n-6 361.5 mg/day, methylphenidate 0.7–1 mg/kg/day,	Improved symptoms.
	Behdani (2013)	[389]	33 controls and 36 interventions	8 weeks	EPA 720 mg/day, DHA 480 mg/day, methylphenidate 1 mg/kg/day	No significant effect.
	Dashti (2014)	[398]	28 controls and 28 interventions	3 days	n-3 1 g/day, methylphenidate 0.9–3 mg/kg/day	Improved symptoms.
	Dubnov-Raz (2014)	[390]	Nine controls and eight interventions	8 weeks	ALA 1 g/day	No significant effect.
	Widenhorn-Müller (2014)	[391]	44 controls and 49 interventions	16 weeks	EPA 600 mg/day, DHA 120 mg/day	No significant effect
	Bos (2015)	[392]	19 controlsand 19 interventions	16 weeks	EPA 650 mg/day, DHA 650 mg/day	Improved attention.
	Matsudaira (2015)	[393]	36 controls and 33 interventions	12 weeks	EPA 558 mg/day, DHA 174 mg/day, GLA 60 mg/day, vitamin E 9.6 mg/day	No significant effect.
	Anand (2016)	[394]	25 controls and 25 interventions	4 months	EPA 180 mg/day, DHA 120 mg/day	Improved symptoms in a subgroup.
	Salehi (2016)	[395]	50 controls and 50 interventions	8 weeks	EPA 100–400 mg/day, methylphenidate 0.5–1 mg/kg/day	Improved symptoms.
	Assareh (2017)	[373]	40 patients	10 weeks	EPA 35 mg/day, DHA 241 mg/day, n-6 LC PUFAs 150 mg/day, methylphenidate 1 mg/kg/day	No significant effect.
	Barragán (2017)	[396]	30 controls and 30 interventions	12 months	EPA 558 mg/day, DHA 174 mg/day, GLA 60 mg/day, methylphenidate 0.5–1 mg/kg/day	Improved symptoms.
	Kean (2017)	[397]	58 controls and 54 interventions	14 weeks	EPA 21.9–29.2 mg/day, DHA 16.5–22 mg/day, vitamin E 0.67–0.9 mg/day	Improved hyperactivity and inattention.
	Moghaddam (2017)	[368]	40 patients	8 weeks	EPA 180 mg/day, DHA 120 mg/day, methylphenidate 1 mg/kg/day	Improvements in symptoms.
	Chang (2019)	[270]	44 controls and 48 interventions	12 weeks	EPA 1.2 g/day	Improved attention and vigilance in patients with low levels of EPA only.

## Data Availability

Not applicable.

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
