# Peer review of "Perinatal Dietary Polyunsaturated Fatty Acids in Brain Development, Role in Neurodevelopmental Disorders"

_nutrients, 2021, doi:10.3390/nu13041185_

Round 1

Reviewer 1 Report

The manuscript has been reorganised and improved. However, some outstanding issues remain:

The section 4.2 about sex determination (lines 419-443) still seems out of the scope of the review as it is written now. To make it a better fit, the authors should mention why sex determination would be important for brain development and neurodevelopmental disorders, and/or how it impacts on PUFAs accretion in the brain, and/or PUFA metabolism. 

Line 513-514: It’s still not clear what the author means here by replenished. This issue regarding the origins of microglia was mentioned in the previous comments. It is really important that the authors explain clearly what the origins of these cells are as this manuscript will be used by students and other scientist who might not be familiar with the field of microglia.

Lines 700-702: What is the take home message here. Is there some room for improvement to design such interventions (period, symptoms, age etc?). Some critical insights from the authors would help the reader get a better understanding of the field.

Minor points:

Line 65: space missing

Line 225 : authors might prefer regarding instead of concerning

Line 279-282: it's not clear what this percentages refer to. Maybe just citing the concentration is enough.

Line 309 : authors might prefer where instead of when here

Line 328 : typo/font/police

Line 422: typo/font/police

Line 469: authors might prefer support instead of warrant

Line 472: authors might prefer regarding instead of concerning

Line 532: typo, resolvin not resolving.

Line 544: the authors might prefer starting or have started instead of are started

Line 610-614 : heavy sentence. Authors might consider breaking it down in two parts.

Line 614-615: it's not clear what the authors mean here. Does it relate to the duration of breast-feeding with supplementation or in general. A sentence clarifying their thought would help.

Line 638 : … (impaired ?) language skills… It feels like one word is missing here.

Lines 706-707: typo font/ police.

Line 716 : if the authors refer to GWAS or other genetic studies then the association is with protein coding genes and not proteins directly.

Line 760-762: this could be explainedpharmacological treatment… This seems controversial as in the previously cited study (from Iran) the standard treatment was also administered.

Line 783 : authors might prefer possibility instead of hypothesis to avoid repetition.

Author Response

The manuscript has been reorganised and improved. However, some outstanding issues remain:

We thank the reviewer for these critical comments, as well as for reviewing our manuscript, together with his/her suggestions of specific sentence modifications. Track changes and alterations are highlighted in yellow in our revised article. Please note that, due to alterations performed within the manuscript, line numbers and sections have now changed throughout. 

The section 4.2 about sex determination (lines 419-443) still seems out of the scope of the review as it is written now. To make it a better fit, the authors should mention why sex determination would be important for brain development and neurodevelopmental disorders, and/or how it impacts on PUFAs accretion in the brain, and/or PUFA metabolism. 

Thank you for critical comments. We have given a rationale for gender differences. We now give references to two pre-clinical studies, using LC n-3 PUFAs and gender differences on the outcomes. Therefore, we have added references, in particular on the preventive effect according to gender of n-3 PUFAs on animal models of neurodevelopmental disorders.  Besides, the two other reviewers found this paragraph very interesting. Thus, we have decided to keep such a section, since some readers would find it interesting.

Line 513-514: It’s still not clear what the author means here by replenished. This issue regarding the origins of microglia was mentioned in the previous comments. It is really important that the authors explain clearly what the origins of these cells are as this manuscript will be used by students and other scientist who might not be familiar with the field of microglia.

Thank you, we have edited this section. It now reads as :

The developmental origins of microglia have been the subject of intense debate [235]. A mesodermal or monocytic origin of microglia has been first hypothesized, until fate-mapping studies showed that microglia derived from erythro-myeloid progenitors from the yolk sac around embryonic day (ED) 7.5 [236]. This primitive haematopoiesis gives rise to pre-macrophages that colonise the whole brain from ED 9.5 and differentiate into microglia from ED 10.5 [237]. Microglia develop according to temporal stages: early, pre and adult, with this final step being reached during the second post-natal week in mice [238].

Lines 700-702: What is the take home message here. Is there some room for improvement to design such interventions (period, symptoms, age etc?). Some critical insights from the authors would help the reader get a better understanding of the field.

We agree with the reviewer that results in RCT studies are sometimes unclear. While some studies show beneficial effects of LC n-3 PUFAs, some show absence of efficacy. We have therefore concluded that “As different risks factors can lead to ASD development, it is quite unclear which treatments or dietary interventions could be efficient in preventing symptoms. However some clinical studies highlight beneficial effects of dietary supplementation on symptoms in young autistic patients.“ 

This has been added to the conclusion of the 5.2 chapter.

Minor points: 

Line 65: space missing

This has been corrected.

Line 225 : authors might prefer regarding instead of concerning

Thank you for your suggestion, the text has been modified following your advice.

Line 279-282: it's not clear what this percentages refer to. Maybe just citing the concentration is enough.

You are right, the % is referencing to the fatty acids composition of milk total lipids. This has been added in the text.

Line 309 : authors might prefer where instead of when here

This has been corrected following your suggestion.

Line 328 : typo/font/police 

This has been edited

Line 422: typo/font/police

This has been edited

Line 469: authors might prefer support instead of warrant

This has been corrected following your suggestion.

Line 472: authors might prefer regarding instead of concerning

Thank you for your suggestion, the text has been modified following your advice.

Line 532: typo, resolvin not resolving.

Thank you, this has been amended.

Line 544: the authors might prefer starting or have started instead of are started

The sentence has been changed following your point.

Line 610-614 : heavy sentence. Authors might consider breaking it down in two parts.

You are right, we split this sentence into two parts.

Line 614-615: it's not clear what the authors mean here. Does it relate to the duration of breast-feeding with supplementation or in general. A sentence clarifying their thought would help.

This sentence relates to breast-feeding in general, without supplementation. It has been clarified in the text.

Line 638 : … (impaired ?) language skills… It feels like one word is missing here.

Indeed, the missing word is now present in the text.

Lines 706-707: typo font/ police.

The police has been edited

Line 716 : if the authors refer to GWAS or other genetic studies then the association is with protein coding genes and not proteins directly.

Thank you for this comment, the association is with protein coding genes, changes have been made in the text.

Line 760-762: this could be explainedpharmacological treatment… This seems controversial as in the previously cited study (from Iran) the standard treatment was also administered.

Thank you, the reviewer is right. Indeed, in the study from Moghaddam et al, MPH was also given, which was also the design in the study from Assareh et al. We have therefore edited the sentence towards: 

However, one clinical trial in ADHD children (6-12 years old) found no beneficial effects of PUFAs supplementation (241 mg/day of DHA, 33 mg/day of EPA and 150 mg/day of n-6 PUFAs) after 10 weeks of treatment [380], when combined with the standard pharmacological treatment (methylphenidate 1 mg/kg/day).

Line 783 : authors might prefer possibility instead of hypothesis to avoid repetition.

This has been corrected, thank you.

Reviewer 2 Report

The revised manuscript "..Perinatal dietary polyunsaturated fatty acids in brain development, role in neurodevelopmental disorders......" incorporated changes in figures and legends with detailed information about the model. The authors added 13 new references in this revised version.

The title is showing that the article restricts the term "dietary" intake of PUFA. However, the article focused on both dietary and endogenous PUFA levels in brain development. The title requires a revision. It could be like "Perinatal polyunsaturated fatty acids in brain development and their roles in neurodevelopmental disorders".

The primary control point for converting dietary PUFA into their longer version is regulated by a set of gene-isoforms of desaturases and elongases. The scope of individual variations in LA and ALA's conversion capacity at population levels varies among races, regions, and dietary practices. This may lead to initiate genetic polymorphisms of desaturase and elongase as a result of irreversible epigenetic pathways over generations. The author may highlight possible hypotheses of the global distribution of desaturase polymorphism concerning race, diets, geography, etc. It may help in targeting the population for supplementation more precisely than exists.

Line 255-256 - The involvement of SREBP has been postulated in the possible mechanism of DHA transport from BBB endothelial cells to neurons.

Table- Highlighting certain rows with different colors can be avoided unless authors want to attract the specific studies.
 The conclusive statement reads a bit generic in particular, and the last sentence is not clearly expressed (Line 854-855)

The consensus outcomes of various clinical trials with n-3 PUFA supplementation and brain development disorders like ADHD, ASD etc. can be placed.

Author Response

The revised manuscript "Perinatal dietary polyunsaturated fatty acids in brain development, role in neurodevelopmental disorders......" incorporated changes in figures and legends with detailed information about the model. The authors added 13 new references in this revised version.

We thank the reviewer for these critical comments, as well as for reviewing our manuscript, together with his/her suggestions of specific sentence modifications. Track changes and alterations are highlighted in yellow in our revised article. Please note that, due to alterations performed within the manuscript, line numbers and sections have now changed throughout. 

The title is showing that the article restricts the term "dietary" intake of PUFA. However, the article focused on both dietary and endogenous PUFA levels in brain development. The title requires a revision. It could be like "Perinatal polyunsaturated fatty acids in brain development and their roles in neurodevelopmental disorders".

The reviewer is right, we have taken this suggestion on board. The title has been edited to reflect this suggestion.

The primary control point for converting dietary PUFA into their longer version is regulated by a set of gene-isoforms of desaturases and elongases. The scope of individual variations in LA and ALA's conversion capacity at population levels varies among races, regions, and dietary practices. This may lead to initiate genetic polymorphisms of desaturase and elongase as a result of irreversible epigenetic pathways over generations. The author may highlight possible hypotheses of the global distribution of desaturase polymorphism concerning race, diets, geography, etc. It may help in targeting the population for supplementation more precisely than exists.

Thank you for your interesting comment. This is a very interesting issue and we have therefore added the following paragraphs to our revised manuscript:

     “Until recently, the capacity of metabolization of PUFA precursors into LC-PUFAs was believed to be uniform in individuals and populations. The discovery that European and African populations carry different forms of fads alleles may partially explain the differences between blood levels of LC-PUFAs in these populations (reviewed in [25]). The geographical differences in fads alleles are probably link to specific selection in European and African population, due to different food habits [26]. This knowledge, in combination of dietary evaluation, may help to refine dietary recommendation target more precisely population for personalized dietary supplementation in pregnant/lactating women and children at risk of altered level of LC-PUFAs.”

Line 255-256 - The involvement of SREBP has been postulated in the possible mechanism of DHA transport from BBB endothelial cells to neurons.

Thank you for this comment. We have added a paragraph in our revised manuscript:

“Thanks to this KO in mice, the authors demonstrated that the expression of Mfsd2a is regulated by sterol regulatory element-binding proteins (Srebps) [94]. Srebp, a transcription factor, exists in three isoforms: Srebp-1a and -1c, both regulating genes required for lipogenesis, and Srebp-2, regulating genes in the metabolism of cholesterol, with the two isoforms -1c and -2 being predominant in the brain [102]. A high level of EPA/DHA in the diet in mice induces a decrease in the expression of Srebp-1 in the brain [103]. Srebp-1 expression is significantly reduced in the dysbindin-1 KO mouse model of schizophrenia and post-mortem brain tissue from patients with schizophrenia [104]. Recently, a team showed in mice that KO of Srebp-1c induced an alteration of GABAergic transmission, leading to symptoms similar to schizophrenia: hyperactivity, depression-like symptoms and social deficits [105]. All these results show that SREBP-1 could play a role in synaptic plasticity and transmission via the regulatory loop with n-3 PUFAs.”

Table- Highlighting certain rows with different colors can be avoided unless authors want to attract the specific studies.

Thank you. Highlights have been removed from all tables.

 The conclusive statement reads a bit generic in particular, and the last sentence is not clearly expressed (Line 854-855)

We have changed the conclusion as following:

“The understanding of the mechanisms underlying perinatal dietary PUFAs to physiological and pathological brain development remains incomplete. However, significant advancements have been made in recent years, with more understanding of how low levels of LC-PUFAs (nutritional and/or genetic) during pregnancy and infancy contribute to the etiopathology of neurodevelopmental disorders. This knowledge is essential to design appropriate nutritional intervention with LC PUFAs in mothers and children with low levels of LC-PUFAs, children at risk of impaired neurodevelopment or women facing at-risk pregnancies. A more comprehensive understanding of the genetic, physiological and behavioural modulators of EPA and DHA status and response to intervention is needed to allow refinement of current dietary LC n-3 PUFA recommendations and stratification of advice to “vulnerable” and responsive subgroups. Overall, PUFA-based gene-diet interactions in humans should provide solid scientific basis for the development of personalized nutritional intervention early in life.”

The consensus outcomes of various clinical trials with n-3 PUFA supplementation and brain development disorders like ADHD, ASD etc. can be placed.

Thank you, we have now added a general conclusion on interventional studies using LC n-3 PUFAs in patients with ASD or ADHD. It now reads as:

     “Interventions using LC n-3 PUFAs in patients with neurodevelopmental disorders produced mitigated results. Indeed, in patients with ASD, 11 out of 18 interventions produced beneficial effects (Table 2), while 21 out of 31 interventions produced beneficial/minimal effects in patients with ADHD (Table 3).”

Reviewer 3 Report

The authors conducted a literature review. The aim of this paper was to discuss the effects of perinatal dietary LC-PUFAs in brain development and their role in neurodevelopmental disorders. This topic is predictably of interest to early life development and nutrition research scientists, and presumably a subject of future discussions. The paper is generally well-written. However, I have some questions and concerns about the manuscript.

1. The word “discuss” is repetitive (lines 15, 17 and 21). Please change it.

2. Lines 41-42: This sentence is confusing.

3. Figure 1: Has poor quality. Please improve it. Arachidonic acid is expresed as ARA in the manuscript while in Figure 1 is expresed as AA. Please change it. Please define list of abbreviations on figure footnotes.

4. Line 328: please change the letter size type in “However”

5. Lines 341-342: “Finally, a recent review also reports mixed effects of LC PUFAs on cognition in preterm children”.

Neuropsychological assessment during infancy and childhood is frequently implemented as a result in nutritional research due to the constant link between some nutrients with brain development and neurocognitive function. However, these abilities are not easy to measure even using validated tests, and it is important to select the most appropriate tests or procedures adapted to the stage of brain development in order to obtain meaningful results. Neurodevelopmental assessment is very limited and it is difficult to obtain significant differences in healthy term infants with available neuropsychological tests and procedures.

The authors expresed the results only in preterm infants, but, what about healthy-term infants?

6. Section 3.2. Gender differences in brain PUFAs accretion and effect of PUFA son sexdetermination.

Very interesting topic. In fact, the adult male brain has ≈10% greater volume than the adult female brain, but lower cortical thickness and white matter tract complexity. Studies performed in children also show gender-based effects on brain structure, suggesting that boys seem to have greater variability in brain structure compared to girls. However, other factors or educational experiences, but not gender, seem to be decisive in brain-cognitive differences between boys and girls. Boys and girls responded differently to environmental stimuli, showing different growth trajectories. Furthermore, children might have different nutritional and hormonal requirements according to their sex. In fact, there is evidence on the specificity of breast milk composition according to the sex of the baby, but studies are still limited and conflicting. Nevertheless, further studies should be carried out to clarify gender effects on brain structure and cognitive performance.

7. Line 422: please change the letter size type in “essential fatty acids, EFA”

8. Section 4.1. Role of PUFAs in synaptogenesis and neuronal development:  

Interesting topic. The evidence so far indicates the need to supplement infant formulas with both fatty acids (DHA and ARA), since studies carried out with DHA without ARA have determined a decrease of more than 50% of ARA in plasma, with respect to the concentrations found in children fed with formulas supplemented with DHA and ARA. However, different systematic reviews and meta-analyzes of published studies have concluded that there is not enough scientific evidence of this association.

9. Line 489: Please change “hippocampi” for hippocampus

10. Figure 2: Footnote: Does not describe the figure correctly, in fact, in the figure does not appear several risk factors, only inadequate levels of n-3 PUFAs.

11. Line 580: “Both genetic alteration and inadequate nutrition....” Among others...

12. Lines 607-613: Repetitive. Please consider rewriting.

13. Section 5.1. Early-life n-3 PUFAs and cognition in infants.

You should consider to include ratio AA/DHA.

14. Section 5.2. Autism Spectrum Disorders (ASD). Table 2: simple size is too small in all studies discussed. It is difficult to draw conclusions.

15. Lines 706-707: please change the letter size type in “Diagnostic and Statistical Manual of Mental Disorders”

16. Lines 750, 752: Repetitive “Similar”

18. Lines 840, 842: Repetitive “Appears”

19. Please include a Methodology section, with data screening, information sources, search components…

20. A risk of bias analysis was not carried out. The authors should consider this analysis for each individual study as well as present a summary of risk of bias analysis in the manuscript (in table, in text…)

Author Response

The authors conducted a literature review. The aim of this paper was to discuss the effects of perinatal dietary LC-PUFAs in brain development and their role in neurodevelopmental disorders. This topic is predictably of interest to early life development and nutrition research scientists, and presumably a subject of future discussions. The paper is generally well-written. However, I have some questions and concerns about the manuscript.

 We thank the reviewer for these suggestions and critical comments on our review. Track changes and alterations are highlighted in yellow in our revised article. Please note that, due to alterations performed within the manuscript, line numbers and sections have now changed throughout.

  1. The word “discuss” is repetitive (lines 15, 17 and 21). Please change it.

Thank you for pointing that out, we have changed it.

  1. Lines 41-42: This sentence is confusing.

There is a differential accretion in the brain between DHA and ARA (accretion) when compared with EPA (negligeable accretion).

  1. Figure 1: Has poor quality. Please improve it. Arachidonic acid is expresed as ARA in the manuscript while in Figure 1 is expresed as AA. Please change it. Please define list of abbreviations on figure footnotes.

Thank you for this comment. It may appear that the version of this figure seen by this reviewer was not the last one that has already been edited. The quality has been improved and arachidonic acid is already expressed as ARA in the manuscript as well as in Figure 1. These comments were raised by reviewers in a previous round of reviewing. However, we agree, a list of abbreviations is now included in the legend of Figure 1, thank you for pointing that out.

  1. Line 328: please change the letter size type in “However”

The police has been edited

  1. Lines 341-342: “Finally, a recent review also reports mixed effects of LC PUFAs on cognition in preterm children”.

Neuropsychological assessment during infancy and childhood is frequently implemented as a result in nutritional research due to the constant link between some nutrients with brain development and neurocognitive function. However, these abilities are not easy to measure even using validated tests, and it is important to select the most appropriate tests or procedures adapted to the stage of brain development in order to obtain meaningful results. Neurodevelopmental assessment is very limited and it is difficult to obtain significant differences in healthy term infants with available neuropsychological tests and procedures.

The authors expresed the results only in preterm infants, but, what about healthy-term infants?

Thank you for this comment. As the reviewer rightly said, neurodevelopment assessment should be performed adequately, in regards to age of the children. The use of validated tests is paramount. This is the reason why we choose to focus especially on preterm birth because most of the studies revealed impairment in cognitive functions of preterm infants and effects of dietary interventions on those functions. Dietary n-3 LC PUFAs supplementation on healthy children is poorly studied and shows mitigated effect on child abilities. However, following your very interesting remark, we choose to add one study conducted on term infants, performed in 2000, that shows interesting effects of formula milk supplementation. It is now written as:

     “A randomized controlled trial conducted on term infants in 2000 revealed that supplementation of formula milk with DHA and AA at an early life stage was improving Mental Development Index of the Bayley Scales at 18 months of age [145]. Nonetheless, it is important to underline that neurodevelopmental assessment should be performed with neuropsychological tests and procedures adapted to the age of children.”

  1. Section 3.2. Gender differences in brain PUFAs accretion and effect of PUFAs on sex determination.

Very interesting topic. In fact, the adult male brain has ≈10% greater volume than the adult female brain, but lower cortical thickness and white matter tract complexity. Studies performed in children also show gender-based effects on brain structure, suggesting that boys seem to have greater variability in brain structure compared to girls. However, other factors or educational experiences, but not gender, seem to be decisive in brain-cognitive differences between boys and girls. Boys and girls responded differently to environmental stimuli, showing different growth trajectories. Furthermore, children might have different nutritional and hormonal requirements according to their sex. In fact, there is evidence on the specificity of breast milk composition according to the sex of the baby, but studies are still limited and conflicting. Nevertheless, further studies should be carried out to clarify gender effects on brain structure and cognitive performance.

We agree with the reviewer that this is an interesting field of study. This was also raised by other reviewers. We have included small paragraphs in regards to gender differences in brain morphology/volume and neurodevelopmental disorders. These sections now read as :

     “In humans, it is known that the brain develops differently depending on gender. In a large human cohort, significant gender differences were observed regarding cortical thickness, fibre organization and total brain volume [159]. These anatomical differences could explain that gender may play a significant role in neurodevelopmental disorders, as observed with autism [160] and attention deficit and hyperactivity disorder [161]. Besides, gender differences are also observed in patients with schizophrenia [162]. However, these differences are sometimes attributed to methodological issues [163,164].”

     “A study on deficiency or supplementation in n-3 PUFA during the perinatal period and for 16 weeks after weaning in mice, shows that the changes in cerebellar FA are more pronounced in offspring females, with a significant effect of diet [175], due to the presence of oestrogen (for review [176]). All these results suggest that ovarian hormones up-regulate DHA content in erythrocytes and brain regions. However, a recent study examining the interaction effects between diet, sex, brain regions, and phospholipid pools in mice demonstrates that DHA concentration is gender independent, while ARA concentration is partially dependent on sex  [177].

            Gender may influence the preventive effects of a n-3 PUFAs supplemented diet on neurodevelopmental disorders in animal models. In fact, supplementation with n-3 PUFAs in pregnant spontaneously hypertensive rat dams (SHR) induces a reduction in hyperactivity and impulsivity in the male offspring, but with no effect, or even opposite effects, in the female offspring [178]. A recent study conducted in a two-hit model, in mice, shows sex-specific preventive effects of LC n-3 PUFAs [179].

            All these studies show that there is a sex effect on the endogenous formation of LC PUFAs. However, further studies are needed to understand the effect of gender, and therefore hormones, on the accretion of PUFAs in the brain.”

  1. Line 422: please change the letter size type in “essential fatty acids, EFA”

The police has been edited

  1. Section 4.1. Role of PUFAs in synaptogenesis and neuronal development:  

Interesting topic. The evidence so far indicates the need to supplement infant formulas with both fatty acids (DHA and ARA), since studies carried out with DHA without ARA have determined a decrease of more than 50% of ARA in plasma, with respect to the concentrations found in children fed with formulas supplemented with DHA and ARA. However, different systematic reviews and meta-analyzes of published studies have concluded that there is not enough scientific evidence of this association.

We would like to thank the reviewer for his insightful comment. 

In section 2.2.2, we added the following paragraph:

     “Interestingly, several recent works report the presence of free and esterified ARA, EPA and DHA-derived oxylipins in human milk [123–125]. Of note, the most abundant form of oxylipins are the ones derived from LA [124], which have been recently reported to be key to brain development in rats [126]. However, whether milk oxylipins play a role as signalling molecules for infant brain development is poorly known.”

In section 4.1, we added the following paragraph:

     “Overall, the mechanistic data brought by animal studies further support the need of LC PUFA for optimal brain development. Concerning post-natal periods, breast milk remains the gold standard, as it provides both ARA and DHA [223]. Concerning infant formulae, the amount of ARA and DHA to be added is still a matter of debate, as recently reviewed [224]. The recent European recommendation does not pinpoint the need of ARA addition in infant formulae, while adding DHA is mandatory (reviewed in [224]). The lack of recommendation on ARA in formulae has been pinpointed by several experts as being a potential risk for improper brain development [225–227], as a diet poor in ARA causes its decreased bioavailability for the brain [228]. Further studies are therefore needed to better understand the mechanisms underlying ARA and DHA association on brain developmental processes [229].”

  1. Line 489: Please change “hippocampi” for hippocampus

This has been corrected.

  1. Figure 2: Footnote: Does not describe the figure correctly, in fact, in the figure does not appear several risk factors, only inadequate levels of n-3 PUFAs.

Thank you for pointing this out. The caption of Figure 2 has been shortened to reflect this.

  1. Line 580: “Both genetic alteration and inadequate nutrition....” Among others…

We totally agree, thank you. The sentence now reads as :

“Both genetic alteration and inadequate nutrition (amongst others), [...] ”.

  1. Lines 607-613: Repetitive. Please consider rewriting.

This has been edited, thank you.

  1. Section 5.1. Early-life n-3 PUFAs and cognition in infants.

You should consider to include ratio AA/DHA.

Thank you, we have added the following sentence at the beginning of section 5.1:

“Explanations have been focused on the balance (ratio) between DHA and ARA, as emphasised before [263–265].”

  1. Section 5.2. Autism Spectrum Disorders (ASD). Table 2: simple size is too small in all studies discussed. It is difficult to draw conclusions.

We totally agree with the reviewer. This is now mentioned at the end of the section on ASD, as well as in the general conclusion of this review :

To conclude, the effects of LC n-3 PUFAs in clinical trials on ASD have led to mitigating effects. As different risks factors can lead to ASD development, it is quite unclear which treatments or dietary interventions could be efficient in preventing symptoms. However, some clinical studies highlight a beneficial effect of dietary supplementation on symptoms in young autistic patients. 

Interventions using LC n-3 PUFAs in patients with neurodevelopmental disorders produced mitigated results. Indeed, in patients with ASD, 11 out of 18 interventions produced beneficial effects (Table 2), while 21 out of 31 interventions produced beneficial/minimal effects in patients with ADHD (Table 3).

  1. Lines 706-707: please change the letter size type in “Diagnostic and Statistical Manual of Mental Disorders”

The police has been edited

  1. Lines 750, 752: Repetitive “Similar”

This has been corrected.

  1. Lines 840, 842: Repetitive “Appears”

This has been corrected.

  1. Please include a Methodology section, with data screening, information sources, search components…
  2. A risk of bias analysis was not carried out. The authors should consider this analysis for each individual study as well as present a summary of risk of bias analysis in the manuscript (in table, in text…)

We thank you for your suggestions (comments 19 and 20). In fact, we did not perform an exhaustive review of all possible interventional studies using LC n-3 PUFAs in patients with ASD, ADHD or schizophrenia. Therefore, our study was not designed as a meta-analysis. Hence, the absence of bias analysis and descriptive methodology. Besides, such meta-analyses have been conducted previously, some of which are cited in the present manuscript. Indeed, for n-3 PUFA interventions in ASD, a meta-analysis was conducted by Mazahery et al, in 2017. For ADHD, a meta-analysis was performed by Chang et al, in 2018. Both of these studies are now cited in the updated manuscript.

While it would be very interesting to conduct a thorough investigation, using meta-analysis, this would be outside of the scope of our current study. We have therefore included some, but not all, relevant studies in the present manuscript, together with non-exhaustive Tables (Table 2 and Table 3). 

 Please note that some references have been added, especially within Table 2. Besides, all of the studies included in previously published meta-analyses (for ASD Mazahery et al 2017 and for ADHD Chang et al, 2018) are indeed included in our 2 tables (respectively Table 2 and Table 3). Great heterogeneity in interventional studies are observed (interventions, duration, sample size and measured outcomes, etc), therefore not all are included in previously published meta-analyses. Hence, we summarised several studies with n-3 PUFAs interventions, some of which were not included in previous meta-analyses. However, one can only be too cautious about the exhaustiveness of such tables.

This manuscript is a resubmission of an earlier submission. The following is a list of the peer review reports and author responses from that submission.

Round 1

Reviewer 1 Report

This manuscript by Martinat et al aims to review the literature about dietary polyunsaturated fatty acids during development and their potential role in neurodevelopmental disorders. The topic is timely and interesting. The manuscript provides a global overview of research on human and rodent models. Knowledge gaps and future directions are nicely highlighted, particularly regarding the cell specific role of PUFAs and the impact/link with its maternal supply. Once refined it will be largely beneficial to the field.

  1. One important limit is that some sections are either not well placed in the manuscript or do not really fit in the topic as it is written. For example, section 4.1 promises to discuss “Differences in PUFAs accretion and activities according to gender”. Unfortunately, the authors fall short of providing any evidence for such differences and instead discuss the impact of PUFAs on sex determination, which in itself is an interested topic but does not explain differences in brain PUFAs according to gender.

Other examples are:

Line 103-106. It is hard to understand the relevance of this section here. The authors discuss a link between SNPs and cytokine levels while they never mentioned the role of PUFAs in inflammation before. It might better suited in a later paragraph of the review.

Line 355-363. The section “several SNPs… decrease of DHA plasma levels” does not seem to be related to the “in situ production of DHA”. While the information provided here is interesting it might be better suited in another section/chapter.

  1. Another issue is the sometimes-contradictory statements found all along the manuscript which does not help determine if this is still contradictory and needs more investigations or if it has been settled. Some examples below:

Line 52-53. “myelination active … sudden start at 32 weeks”. As written this sentence is confusing and first 2 years of life seems contradictory to 32 weeks of gestation. 

Line 92-95. It’s not clear what the authors try to state here. Are very LC PUFAs important for the brain or not? They first emphasize that Elovl4 is found in the brain but then say that its products are present only in traces. Does it mean that Elovl4 as another role in the brain?

Line 175. The statement “during postnatal life, only little LC  PUFAs accretion is observed” is not consistent with the previous one Line 169 where it is stated that “DHA accumulation is massive… in the first two years of life”. If these are contradictory findings the authors should clearly state that this is controversial.

Line 372-375. The sentences in this section “indeed, numerous studies… DHA in women the in men.” Are actually contradictory to the first statement of this chapter about gender differences (“there are very few studies in the differential status”). The authors should clarify what it is they exactly mean.

  1. In the chapter 5 about mechanisms of neurodevelopment, only synaptogenesis is discussed leaving out all the previous steps (neurogenesis, cell migration etc) of brain development. If this is a deliberate choice from the authors they should justify it.

  1. As a suggestion, the organisation of the sections in the chapter about neurodevelopmental disorders should be similar as much as possible. The way it is written for schizophrenia (human evidence, animal models, intervention studies in human) feels easier to read.

Other points and suggestions:

Line 49. Limiting postnatal development to myelination is reductionist. Synaptogenesis, synaptic pruning, cell death occur during this period. In particular as the authors will focus on synaptogenesis later in the review they might want to highlight it here instead of myelination.

Line 50. “with anatomical and processes” is unclear.

Line 77. Reference Formatting.

Line 163-166. The authors explain the differences between grey and white matter by the enrichment of DHA in synaptosomes. Could lipid composition of myelin also contribute to this difference?

Line 172. “while” is repeated twice. Consider editing.

Line 222-224. These are very interesting studies. The authors could elaborate a bit. Is p-FABPpm the only transporter for ARA and DHA?  Is it known what is the cause of this difference in affinity between foetal and maternal compartment? Is it specific to PUFAs?

Line 249. Reference Formatting.

Line 250. What are partial or total mutations? Do the authors mean point mutation vs deletion?

Line 323. Typo Gould et alia

Line 340. Consider “the authors showed” instead of “they have shown”

Line 343-345. The authors write that D5D and D6D have been cloned. As D5D and D6D are the proteins’ name, gene names (FADS1 and FADS2) would be more appropriate in that case.

Line 384. The author might mean “a greater proportion” instead of composition.

Line 457. The current view in microglia development is that they are a unique population of macrophages and solely derived from the first wave of hematopoiesis from the yolk sac. The authors should definitely correct this.

Line 513. References are needed for this statement.

Line 525. The expression middle age is confusing. What period do the authors exactly refer to?

Line 574-575. Repeat of “one study”. Consider rephrasing.

Line 604. “exposed” instead of “exposure”.

Line 632. Low birth weight and premature birth are not really environmental factors but rather possible consequences of the impact of such factors on pregnancy.  

Line 633. Genetic studies cannot identify associations with proteins but rather with genes/DNA sequences encoding proteins.

Line 647. “was able to alleviate” or “alleviated” instead of “was able to alleviated”.

Lines 722-724. Can the dysfunctions of the dopaminergic system seen in Schizophrenia models also apply to ADHD?

Reviewer 2 Report

The review article "Perinatal dietary polyunsaturated fatty acids from the placenta to brain development, a role in neurodevelopmental disorders" utilized data and consolidated information extensively. The manuscript has several dimensions that are elaborated adequately. However, in some sections, the article should be tightened. The figure work should be revised.

The title is not justified using the term "placenta" in this manuscript. It could be misleading if readers expect information about the placenta's role in neurodevelopmental disorders as very few citations are available (out of 399 references) with placenta, LCPUFAs, and fetal development.

Fig.1 Variation in SNPs is associated with genes responsible for endogenous conversion of LA and ALA to their longer version of respective LCPUFAs. LCPUFAs represented as a circle and bold should not include LA and ALA as these are not LCPUFAs.

Fig.2 this figure must be revised to define with discrete information. Is the model is hypothetical or driven by data? It is not clear the context where Figure 2 is placed in this manuscript. It is not clear from the figure about the causes of adverse events. The author may define each causality with citations in the model. If SNPs are adverse events, specify those by pointing out the stages (mating/birth/weanling), name of the SNPs( rs..?) along with citations, etc. SNPs and n-3 deficiency are different events. The first one is genetic, and the second one is environmental (diet). These two events should be labeled in different colors. The term "total brain" is not precise. Does it mean total PUFA content in the brain? "Alteration" is a generic term, was used almost everywhere in this figure. Can it be specified in some cases?

In many places (eg. line 273), comma (,) has been used in place of decimal (.). This may create confusion with "text comma." It could be due to the word processor's default keyboard setting in some countries (based on my own experience before!).

Section line 202-304..there are too many paragraphs within a similar context.

Line 202- the tile of the sections does not reflect the text precisely. This section (line 202-332) can be divided into areas with specific titles e.g. breast milk, LCPUFAs and brain development etc.

The supplementation of n-3LCPUFAs in the case of pre-term in particular reference with cognitive development is not reported well.

Line 332- "in situ" production can be replaced with "endogenous" production as the latter one is more appropriate to define the condition.

Line 415-443- the sections are loosely written. Too many paragraphs..

Line 81- Can it be a recommendation of n-3LCPUFA intake regarding the amount required for optimal brain development across the globe universally regarding demanding brain development?

Line 108-109 while there is no data available if low DHA status in vegetarian pregnant women affects children's cognitive development, however, are clinical data available on excess LA rich diet affect cognitive development (neuroinflammation)?

Arachidonic acid can be labelled uniformly as "ARA" instead of AA.

Single nucleotide polymorphism "SNIP" or "SNP" should be placed uniformly in the manuscript.

The conclusive statement should be revised with precise information.  The toxic stress effects during pregnancy and its long-term consequences on locomotors and mental development should be highlighted.

Reviewer 3 Report

Authors made a good effort to discuss the importance of accretion of LC-PUFA in the developing brain, highlighting the existing literature gaps. Authors also discuss the vital role of maternal dietary intake during the periods of gestation and lactation, as well as the implication of the placenta and the maternal milk in LC-PUFA supply to the offspring. The effects of gene polymorphisms on brain LC-PUFA accumulation are also described. In case of impairment of this accretion, authors provide a thorough discussion on potential outcomes regarding neurodevelopmental diseases and psychosis.

However, the manuscript needs extensive English editing; too long sentences are used and some points in the text are not easy to understand. The same abbreviations should be used throughout the manuscript and figures should be mentioned accordingly in the text. The introduction section is too long and restructuring is advised. Additionally, authors should avoid repetitions throughout the manuscript. 

Introduction: Authors highlight the issues to be discussed; however, the introduction is too long and somewhat incoherent. Authors should make a brief introduction describing the lipids (and related genes) that are vital for brain development, their origin (dietary or synthesized), as well as their impact on neurodevelopmental disorders later in life.

Lines 25-27: please add a reference.

Lines26-27: please change to: ….”which have long-term outcomes for brain health that range from neuropsychiatric diseases to cognitive decline and neurodegenerative disorders.”

Lines 27-28: please add a reference

Lines 28-29: please add a reference

Line 46: please change to “….maternal nutrition intake influences….”

Line 47: please change to “emphasized on….”

Line 48: please change “….for brain growth”

Line 55: please define receptors NMDA, AMPA, as well as neurotransmitter GABA used for the 1st time in the text.

Line 59: The phrase does not make sense “Nutrition is susceptible to influence….”, did you mean: “Nutrition can influence….”.

Lines 63-65: Too long sentence - does not make sense. Please rewrite, perhaps “Nowadays,there are nutritional formulas designed for food insecure mothers and infants to promote normal metabolic function and weight gain. However,data on nutritional strategies supporting optimal brain development are limited”.

Line 68: “….are associated….”.

Lines 69-70: “…considered as crucial for brain development also relying on dietary supply, are….”

Line 74: “….ALA is found…, while LA is found….”

Lines 71-75: add reference

Lines 75-79: the word “respectively” is used twice. It is suggested to rephrase, perhaps “Once consumed, both ALA and LA are metabolized into long-chain (LC) PUFAs; ALA to eicosapentaenoic (EPA, n-3) and docosahexaenoic acid (DHA, n-3), and LA to arachidonic acid (ARA, n-6).

Line 79: please change the format of the reference.

Lines 81-84: too long sentence. Please rewrite e.g. “Of note, health organizations recommend to increase the consumption of n-3 LC PUFAs-rich marine products. Nevertheless, the origin, safety and sustainable supply of the marine sources have raised concerns. Alternative sustainable sources, such as algae-based n-3 PUFAs, are investigated”.

Line 85: please remove the parenthesis.

Line 85: please rewrite “Conversion of ….. to LC PUFAs”

Line 86: “compete”

Lines 88-90: “delta-5 desaturase (D5D) and delta-6 desaturase (D6D) are encoded by fatty acid desaturase 1 (fads1) and fatty acid desaturase 2 (fads2) respectively, while elongase 2 (Elovl2) and elongase 5 (Elovl5) are respectively encoded by elovl2 and elovl5.”

Line 91: “The last step for DHA synthesis involves translocation of ….”

Line 94: I think “detected” suits better than “found”

Line 100: what do you mean by “plasmatic”, meaning in plasma or not truthful?

Line 109: remove the phrase “which was reviewed in 2017”, does not make sense.

Figure 1 is fuzzy.

Line 111-114 (figure 1 legend): “Desaturase enzymes [delta-5 desaturase (D5D) and delta-6 desaturase (D6D)]encoded byfatty acid desaturase genes(fads1 and 2 fads2 respectively), as well as elongase enzymes[elongase 2 (Elovl2) and elongase 5 (Elovl5)] encoded by elovl2 and elovl5 genes (respectively), are involved in desaturation and elongation of PUFAs from the n-6 (cyan) and n-3 (orange) pathways”.

Line 116 (figure 1 legend): “plasmatic” or plasma?

Line 120: “….especially in the brain….”

Line 128: please rephrase the sentence e.g. “Their role in brain inflammation has been reviewed recently, however data on the developing brain are rather scarce and are discussed extensively in the present review”.

Figure 2: you don’t refer to figure 2 in the text. The figure looks fuzzy.

Line 135: “.…formed during”….

Line 140: Nicely introduced to the issues to be covered.

Line 142: please change the format of the reference.

Line 148: remove “of” from “discuss of the role”

  1. PUFAs and brain development

Lines 163-165: change to “These differences could be due to the accumulation of DHA in synaptosomes and ARA in vascular cell membranes of the brain, which is suggested to serve as a depot for ARA”.

Line 169: please define: “DHA accumulation” where…. in the brain?

Line 170: “In humans, ….”

Line 176: “In humans, ….”

Lines 175-176: why do you refer to rodents?

Line 186: remove the word “level”

Line 191: what do you mean by “its impairment”?

Lines 198-199: please rephrase

What about DHA, EPA and ARA accretion in brain cells of preterm neonates?

  1. Needs of LC PUFAs to the pre and post-natal developing brain: transport and dietary maternal supply

This is a very interesting and well-written section.

Lines 204-205: there is no need of repeating. Check the whole manuscript for repetitions.

Line 212: “….advise…”

Lines 211-213: please rephrase e.g. “According to the World Health Organization, pregnant women should consume at least 200 mg/day of DHA.”

Lines 228-229: please check repetition.

Line 265: “prevents” not “offers reductions”

Line 292: does not make sense, please rephrase: ”FA from food sources in lactating mothers can be used in three ways.”

Line 295: “can influence” not “impact”

Line 298: please use the same tense (past or present) throughout the text.

Line 299: please check repetition.

Line 313: DHA is already defined

Line 319:  please check repetition.

Line 330: “transfer”

  1. In situ production of PUFAs in the developing brain

Line 334: rephrase the sentence e.g. “….supply to the infant brain occurs during gestation and lactation….”

Line 337: “…..released in the developing brain….”

Line 340: remove “using such a paradigm” and rephrase e.g. “The results showed that the postnatal….

Line 346: I suggest adding the “Elovl” in the parenthesis, i.e. (Elovl 1, 3, 4, 5 and 6)

Line 350: use the same abbreviations throughout the manuscript

Line 362: define DPA for the first time used in the text

Lines 355-365: use the same abbreviations throughout the manuscript

Line 360: “….plasma phospholipids….”

Line 362: “….more likely to induce….”

Line 369: “….adult women than men.”

Line 381: “A recent study in rats….”

Line 400: please remove “….due to the rise of prostaglandins…..”.

Line 407: “…oocytes that mature….”

  1. Mechanisms of action of PUFAs on neurodevelopment

Line 425: define “BDNF”

Line 432: what supplementation?

Line 449: “….regulate….”

Line 451: remove “including” and “which will be further described”

Line 445-452: please check repetition.

Line 454: “…..their role….”

Line 461: “….influences microglia….”

Line 462: “….increasing the risk for neurodevelopmental disorders” suits better, remove “therefore”.

Line 469: remove “from” or add the type of study.

Line 479: “....promotes…”

Line 482: please rephrase, e.g.  “contributes to” Instead of “at the origin”

Line 483: remove “of the addition”, do you mean “both”?

Line 504: use “Factors contributing to” instead of “The etiology”

Line 505: “As an example, clinical and pre-clinical studies show that….”

Line 511: “….playing a key role”

Line 527: “Due to the observed positive association between n-3 PUFA serum levels and cognitive abilities in infants, there have been various attempts…, as well as in young patients with autism….”

Line 539: add “….positive correlation”,

Line 541: what “term”?

Line 554: “…..a n-3 PUFAs deficient diet….”

Line 569: “….for ASD development”, “With regard to PUFAs,….”

Line 576: you mean “low dietary PUFAs intake”?

Line 588: “in meta-analyses” not “using”

Line 589: “….in preterm born toddlers presenting ASD….”

Line 590: there is no need to explain what LA and ALA are in the parenthesis

Line 593: remove one “trials”  

Line 604: please rephrase, e.g. “In another study, in which mice were exposed to….( intraperitoneal injection of lipopolysaccharide at E17….)”

Line 625: please define DSM-V used for the first time in the text.

Lines 649-650: “Similar results were found by another study, in which 20-25 mg/kg/day of EPA and 8.5-10.5 mg/kg/day of DHA were consumed for 16 weeks, while earlier studies revealed mild improvements of such treatment”.

Line 658: a meta-analysis is not used, please rephrase e.g. “in a meta-analysis on 699 children with ADHD from 10 clinical trials…”

Lines 671-672: please rephrase, perhaps “To conclude, effects of n-3 LC PUFAs supplementation on ADHD patients are ambiguous.”

Line 680: “…on spontaneous…”

Lines 685-688: use the same tense, e.g. past tense throughout the text

Lines 694-697: “…..neurodevelopmental insult….inflammatory…..infectious….., known as the two-hit hypothesis,…..”

Line 697: remove “several”

Line 753: do you mean “plasma levels”?